# Structural dynamics in the evolution of SARS-CoV-2 spike glycoprotein

Valeria Calvaresi [1] ✉, Antoni G. Wrobel [2] ✉, Joanna Toporowska[1], Dietmar Hammerschmid [1], Katie J. Doores [3], Richard T. Bradshaw [1], Ricardo B. Parsons [1], Donald J. Benton [2], Chloë Roustan[4], Eamonn Reading [1], Michael H. Malim [3], Steve J. Gamblin [2] & Argyris Politis [1,5,6] ✉

SARS-CoV-2 spike glycoprotein mediates receptor binding and subsequent membrane fusion. It exists in a range of conformations, including a closed state unable to bind the ACE2 receptor, and an open state that does so but displays more exposed antigenic surface. Spikes of variants of concern (VOCs) acquired amino acid changes linked to increased virulence and immune evasion. Here, using HDX-MS, we identified changes in spike dynamics that we associate with the transition from closed to open conformations, to ACE2 binding, and to specific mutations in VOCs. We show that the RBD-associated subdomain plays a role in spike opening, whereas the NTD acts as a hotspot of conformational divergence of VOC spikes driving immune evasion. Alpha, beta and delta spikes assume predominantly open conformations and ACE2 binding increases the dynamics of their core helices, priming spikes for fusion. Conversely, substitutions in omicron spike lead to predominantly closed conformations, presumably enabling it to escape antibodies. At the same time, its core helices show characteristics of being pre-primed for fusion even in the absence of ACE2. These data inform on SARS-CoV-2 evolution and omicron variant emergence.

Severe acute respiratory syndrome coronavirus 2 (SARS-CoV-2) is the pathogen responsible for the ongoing COVID-19 pandemic[1]. To bind to and fuse with the host cell, SARS-CoV-2 utilizes its transmembrane glycoprotein spike[2], which binds to the human receptor angiotensin-converting enzyme 2 (ACE2)[3–6], expressed on the surface of cells in the respiratory tract. Upon receptor binding and protease cleavage, spike undergoes conformational changes that facilitate fusion with the host cell membranes and subsequent release of viral components into the cytosol of the infected cell. Spike is also the antigen of all vaccines currently on the market[7] and elicits a strong immune response in vaccinated[8] and infected individuals[9].

Spike is a large clove-shaped homotrimer with each spike monomer composed of two subunits: S1 and S2. The S1 mediates ACE2 binding and partially covers the S2, which forms a large helical stalk responsible for the membrane fusion event. The S1 subunit comprises the N-terminal domain (NTD), the receptor binding domain (RBD), and the RBD-associated and the NTD-associated subdomains (RBD-s and NTD-s, also called SD1 and SD2 respectively)[3,4]. The S1 and S2 subunits are connected by a flexible disordered linker containing a polybasic protease cleavage site, which can be cleaved by furin[5,10–12]. An additional cleavage site is located in the S2 subunit (S2' site), upstream of the fusion peptide (FP)[12,13].

[1]Department of Chemistry, King's College London, SE1 1DB London, UK. [2]Structural Biology of Disease Processes Laboratory, The Francis Crick Institute, NW1 1AT London, UK. [3]Department of Infectious Diseases, School of Immunology and Microbial Sciences, King's College London, SE1 9RT London, UK. [4]Structural Biology Science Technology Platform, The Francis Crick Institute, NW1 1AT London, UK. [5]Faculty of Biology, Medicine and Health, School of Biological Sciences, The University of Manchester, M13 9PT Manchester, UK. [6]Manchester Institute of Biotechnology, The University of Manchester, M1 7DN Manchester, UK. ✉e-mail: valeria.calvaresi@kcl.ac.uk; antoni.wrobel@crick.ac.uk; argyris.politis@manchester.ac.uk

In the early stages of SARS-CoV-2 infection, spike undergoes several conformational changes[11,14,15]. On the virus surface, spike exists in a metastable prefusion state, where S1 and S2 subunits interact, and the spike trimer transitions between closed and open conformations. Closed conformations minimize the exposed viral antigenic surface, thus allowing spike to escape neutralizing antibodies, but prevent it from binding to ACE2. In open conformations, spike can recognize ACE2, while increasing its exposed antigenic surface. The transitions between the open and closed conformations are associated with the trimer opening and large domain movements, especially of the RBDs, which erect in order to engage the receptor. Upon receptor binding, spike is primed for fusion and, subsequently, undergoes major structural rearrangements: S1 shedding and a major refolding of its S2 core upon fusion[14,15]. The spike opening and fusion are enhanced by proteolytic cleavages by proteases, such as furin, TMPRSS2, and cathepsins[11–13,16,17].

Early in the pandemic, spike acquired the D614G substitution in the NTD-s, which promotes G614 spike to adopt more open, thus receptor-accessible, conformations compared to D614 spike of the original Wuhan isolate[18–21]. The D614G substitution disrupts the salt bridge with K854 in the fusion peptide–proximal region (FPPR) of a neighbouring monomer, which has been proposed to be the crucial region for driving spike transition from the pre- to post-fusion conformation following the receptor binding[18,19]. A structural element in the NTD-s, located in the vicinity of the FPPR of the neighbouring monomer, named the '630 loop' (residues 620–640), becomes ordered in G614 spike, while appearing disordered in Wuhan spike. When structured, the 630 loop can insert into a gap between the NTD and RBD-s of the same protomer, stabilizing the NTD-s[19]. The opening of the RBD in G614 thus correlates with the 630 loop and the FPPR moving away from the positions adopted in Wuhan spike. The 630 loop is also located in proximity of the S1/S2 boundary, suggesting that NTD-s and RBD-s, together with the FPPR, might also undergo conformational rearrangements upon fusion, although their character remain unclear.

The D614G substitution endowed the virus with a fitness advantage and higher transmissibility[22–24], facilitating the acquisition of further mutations. Hence, several SARS-CoV-2 strains developed on the background of the D614G, some of them of increased virulence and classified as variants of concern (VOCs)[25]. VOCs include B.1.1.7 (alpha), B.1.351 (beta), P.1 (gamma), B.1.617.2 (delta), and B.1.1.529 (omicron), each with spike carrying a unique set of mutations yet sharing G614 (Supplementary Fig. 1). Numerous cryo-EM structures of spikes of VOCs have been solved[26–34], providing accurate high-resolution structural information and describing the extent of opening of various variants. However, as dynamic events largely dominate the spike function, cryo-EM structures alone are unable to uncover the structural dynamics underlying spike opening, ACE2 receptor binding and how this primes the exposure of the S2 trimeric core for fusion, in the ancestral spike and in spikes of VOCs. Due to inner challenges associated with the spike size and its conformational heterogeneity, only a handful of studies have probed its structural dynamics to date[35–42]. We thus lack a clear understanding of the perturbation that mutations in VOCs cause on the spike conformational landscape and how they impact spike function and the dynamics of its epitopes[43].

Hydrogen-Deuterium eXchange coupled to Mass Spectrometry (HDX-MS) is a sensitive technique to interrogate the structural dynamics of proteins and their complexes[44,45]. The protein deuterium incorporation over time measured at peptide level by MS informs on local structural dynamics, global conformational changes, and binding events. Here, we carried out HDX-MS to compare the structural dynamics of spike of the original Wuhan isolate and of VOCs previously and currently circulating (alpha, beta, delta and omicron), as well as studying how they interact with ACE2, and the dynamic events

associated with their priming for fusion. We utilized the G614 spike, carrying the dominant D614G substitution shared among all spikes of VOCs, as a reference state to compare the spike variants and their preference for the open (or closed) conformation.

We show that the RBD-s is the subdomain responsible for modulating the spike opening, whereas an increased flexibility in the NTD characterizes the evolutionary divergence of spike of VOCs from original spike and underlies their immune evasion. Furthermore, we demonstrate that spikes of alpha, beta and delta variants assume predominantly open receptor-accessible states and that the binding to ACE2 increases the dynamics of their S2 core helices, priming spikes for fusion. In contrast, spike of omicron variant evolved to favour the closed state, probably to escape immunity, yet it is already primed for fusion even in the absence of ACE2, which likely enables the transition to postfusion state with minimal receptor binding. Taken together, our data shed light into the conformational evolution of the SARS-CoV-2 spike glycoprotein from the original Wuhan isolate to the now dominant omicron variant.

## Results

The experimental results in this work are all based on HDX-MS data, which inform on protein dynamics at the resolution of peptide segments. The full depth of the insights from protein HDX can be best exploited by reference to other structural studies, particularly cryo-EM[46]. The HDX datasets reported here are of apo and ACE2-bound spikes from the original Wuhan virus and D614G, alpha, beta, delta, and omicron variants, studied under the same HDX conditions. It is important to note that these conditions are necessarily different, to a greater or lesser degree, to those used in the cryo-EM studies. To study spike structural dynamics and receptor binding events, we chose to focus solely on the pre-fusion conformations of spike. Hence, in all analyses we used a version of spike ectodomains stabilized by R682S, R685S substitutions in the polybasic cleavage sites and K986P, K987P (2P) substitutions, which combined make the spikes furin-uncleavable and unable to transition to the postfusion conformation[11,47]. The following sections are presented as subsets of HDX-MS data, where those are compared to other structural studies to aid the logical narrative of the manuscript.

### The spike open conformation shows a characteristic HDX profile

Previous structural studies have shown that the D614G substitution in Wuhan spike promotes the open conformations[18–21]. Therefore, to map dynamic events directly linked to trimer opening, we first compared the HDX of G614 spike and Wuhan (D614) spike. We followed the HDX of 308 peptides, spanning approximately 82% of spike sequence (Fig. 1 and Supplementary Fig. 2).

The HDX profile of G614 spike showed increased HDX in regions of the RBD that become solvent exposed and receptor-accessible when spike is in open conformation (PDB: 7BNN[18]) and are occluded or partially occluded when spike is in closed conformation (PDB: 6ZGE[11]), such as RBD regions 407–422 and 456–471 (Fig. 1 and Supplementary Fig. 3). In Wuhan spike, the segment 834–851, spanning the FPPR, showed protection from HDX only at very early time points, indicative of it being highly dynamic, despite its ordered appearance in the structure of Wuhan spike in closed conformation[14,15,18,48] (Fig. 1c). Notably, this segment showed increased HDX in G614 spike, as a consequence of the disruption of the D614-K854 salt bridge, thus appearing even more dynamic compared to Wuhan spike. G614 spike showed respectively decreased and increased HDX in the segments 621–627 and 628–643, which we interpreted as sign of the structural rearrangements in the 630 loop associated with spike transition from closed to open conformations. Furthermore, we saw increased HDX in helices of the S2 stalk (HR1, FP and CH regions), which we associated with loosening of interactions holding the G614 trimer together. This is in line with previous structural studies showing reduced

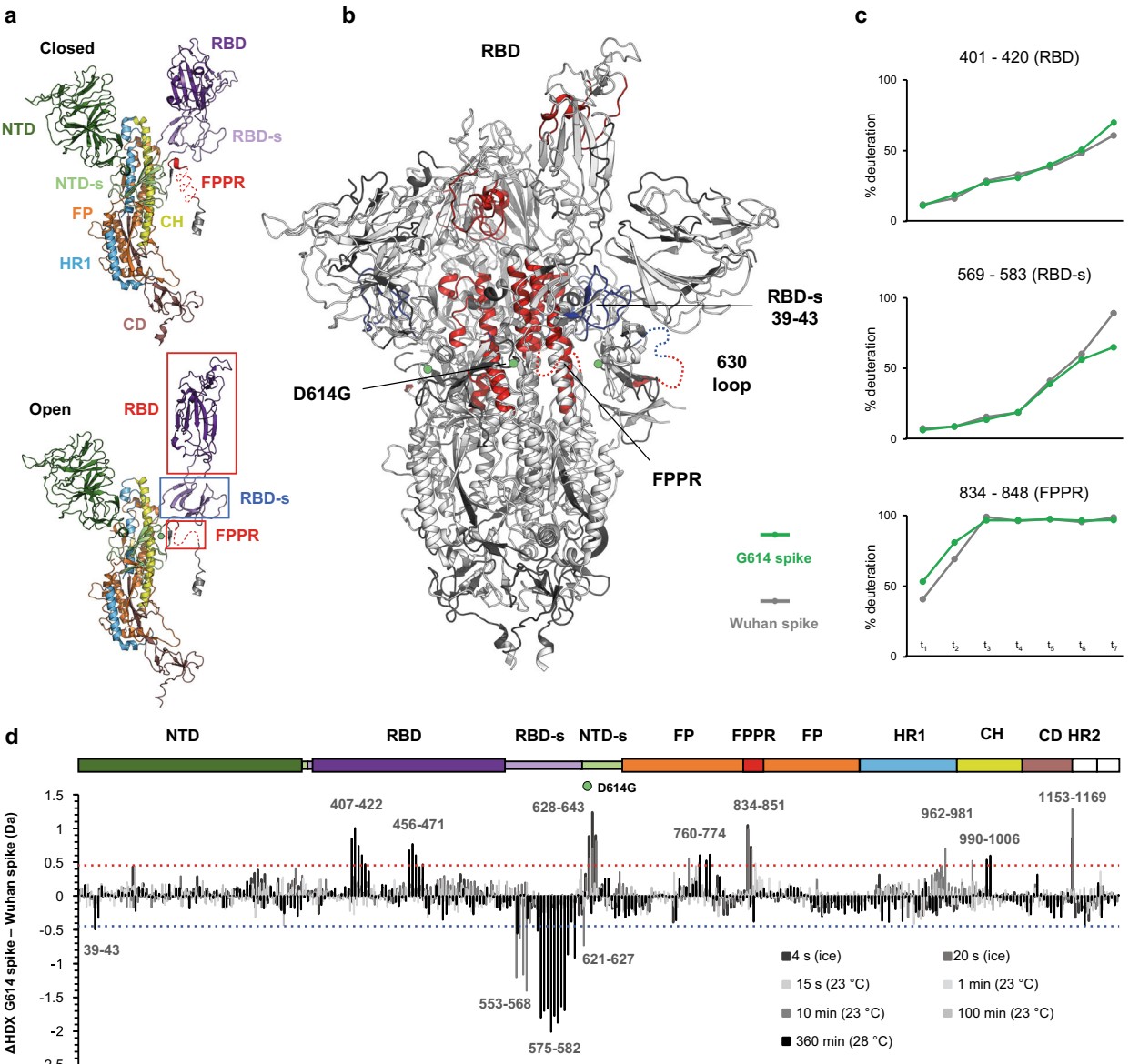

**Fig. 1 | Differences in structural dynamics between Wuhan (D614) spike and G614 spike. a** A single monomer in closed (top) and open (down) conformation is represented (PDB: 7bnn[18]), with individual domains of spike indicated. **b** Regions with differences in HDX are superimposed on the structure of G614 spike with one RBD erected (PDB: 7bnn[18]), coloured in red for increased dynamics and in blue for reduced dynamics. Regions not mapped in the structure and showing HDX effects are inserted as dashed lines in one single protomer. **c** Uptake plots of model peptides of the allosteric axis (FPPR – RBD-s – RBD) that regulates the closed-to-open state transition. **d** Differential plot illustrates the difference in HDX (ΔHDX) between G614 spike and Wuhan spike across the time points studied. The dashed red light indicates the threshold of significance for increased HDX; the dashed blue line indicates the threshold of significance for reduced HDX. The spike domains are illustrated in the top bar and the green sphere is positioned at the level of the D614G substitution.

monomer-monomer interaction area, especially in the closed conformation of the G614 spike when compared with Wuhan spike[18]. Finally, the RBD-s showed markedly lower HDX in G614 spike than in Wuhan spike, which revealed to us that a decrease in dynamics of this subdomain correlates with spike assuming the open conformation.

### Early VOC spikes show increased opening, omicron spike is mostly closed

Having established the HDX hallmarks of the open conformation, we used this knowledge to analyze spike variants, and compared the HDX of spikes of alpha, beta, delta and omicron variants to G614 spike. To the best of our knowledge, this is the first analysis where the spikes of most dominant variants are studied in parallel−under the same sampling conditions−enabling us to understand the putative effect of amino acid changes on their structural dynamics and comprehensively

compare their degree of opening. We followed 275 peptides on average, reaching approximately 80% sequence coverage for each spike variant (Supplementary Figs. 5−8). The impact of differences in chemical exchange constants $(k_{ch})$[49,50] on the HDX of spike variants was examined and deemed negligible for a qualitative assessment of HDX effects (Supplementary Fig. 4 and Supplementary Data 1). We observed the HDX of their RBD, RBD-s, and 630 loop to infer information on their preference for the open or closed state relative to G614 spike.

Alpha, beta and delta spikes displayed increased HDX in the RBD, decreased HDX in the RBD-s and decreased/increased HDX in the two adjacent segments of the 630 loop (Fig. 2 and Supplementary Figs. 9−11) compared to G614 spike, all three indicative of these spike variants showing higher preference for the open conformation. Beta spike showed the highest magnitude of these effects, in agreement with cryo-EM studies that indicated its full transition into the open

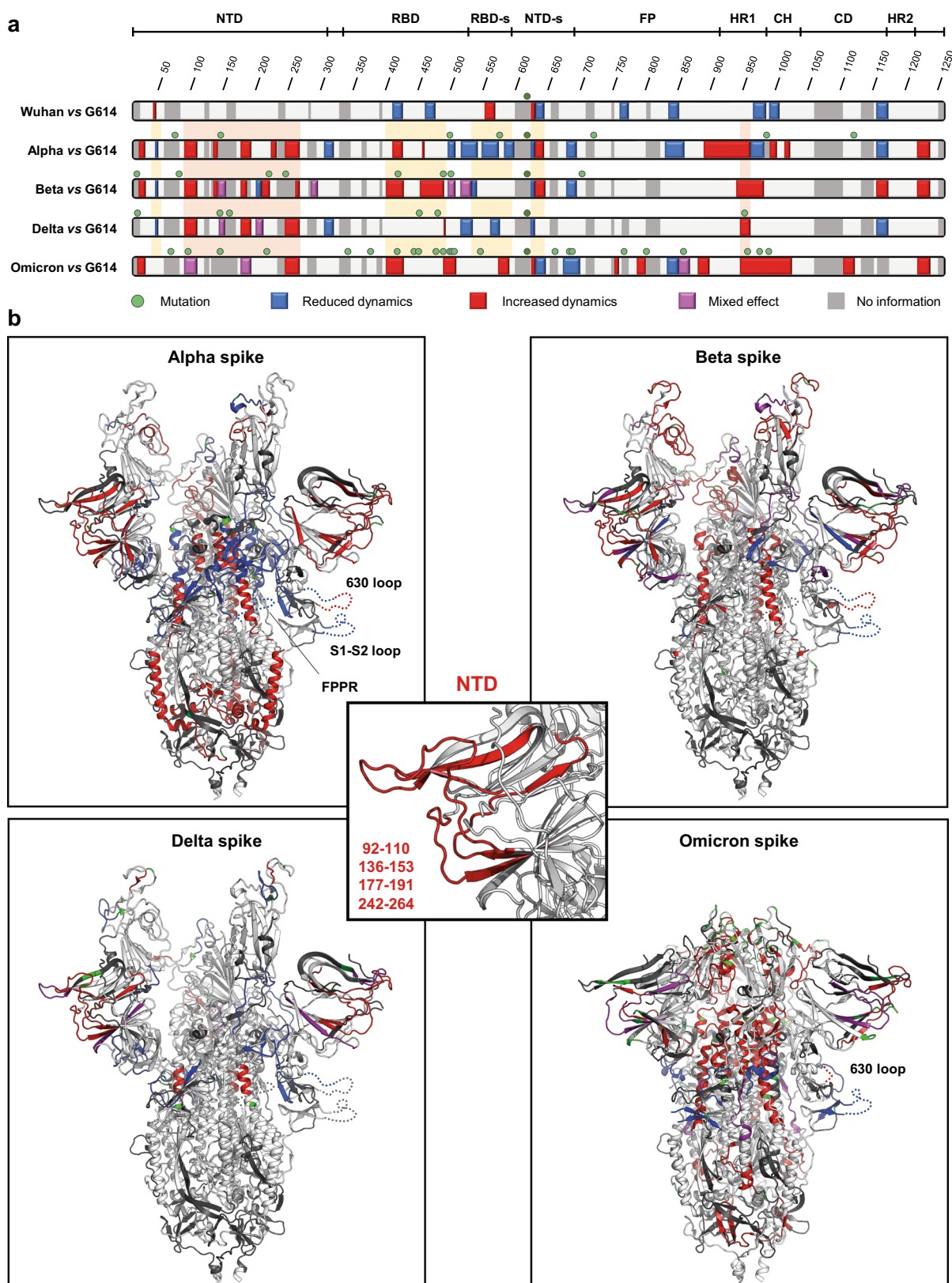

**Fig. 2 | Conformational evolution of spike. a** Map of HDX effects. Regions of various spike trimers with a significant difference in HDX compared to G614 spike are coloured along the spike sequence. In red: increased HDX; in blue: decreased HDX; in magenta: mixed effect; in light grey: no difference in HDX; in dark grey: no information; green spheres: sites of amino acid changes. Yellow rectangles frames regions reporting on a modulation of closed/open state. Orange rectangles frames regions of HDX effects in common among the various spike trimers. **b** HDX effects

are superimposed on the structure of G614 spike with two erect RBDs (PDB: 7bno[18]) for alpha, beta and delta spikes, and on the structure of D614 spike in closed conformation (PDB: 6zge[11]) for omicron spike. Regions not mapped in the structure and showing HDX effects are inserted as dashed lines in one single protomer. Sites with amino acid changes are coloured in green. In the central panel, conformational effects in common among spike variants are superimposed on the NTD subdomain.

state[26]. Specifically, our data showed largely enhanced HDX of the beta spike RBD, especially at the level of the K417N substitution (residues 400–427) (Fig. 2 and Supplementary Fig. 10). As residue 417 is positioned on the RBD-RBD interface of the trimer, N417 presumably confers loose interface packing to close-conformation monomers compared to K417, thus favouring complete opening of beta spike.

In contrast to alpha, beta, and delta spikes, the HDX profile of the 630 loop and RBD-s of omicron spike was similar to that of Wuhan spike (Fig. 2 and Supplementary Fig. 12), indicating it is more closed than G614 spike. However, compared to G614 spike, omicron spike manifested increased HDX at the level of the following substitutions in the RBD: K417N, shared with beta spike, conferred flexibility to segment 400-427, whereas S477N and T478K, the latter shared with delta spike, and E484A, shared with beta spike, conferred flexibility to segments 456–471 and 487–506 (Fig. 2 and Supplementary Fig. 12). These features indicate increased inner flexibility of the RBD of omicron spike, suggesting that it adopts a less compact closed state than Wuhan spike.

Notably, N501Y, shared among alpha, beta and omicron spikes, induced stabilization (decreased HDX) of the receptor binding motif (RBM) in segment 495–506 of alpha and beta spikes (Fig. 2 and Supplementary Figs. 9, 10) but not in omicron spike, likely due to the presence of neighbouring substitutions in the latter (Q493R, G496S, Q498R, and Y505H) that confer increased HDX (Fig. 2 and Supplementary Fig. 12). N501Y has been shown to enhance spike affinity towards ACE2 in alpha and beta spikes, whereas this destabilizing effect we now observed likely explains the previously described limited affinity enhancement in omicron spike[51].

## Spikes of all VOCs show increased dynamics in the NTD and S2

Individual mutations in VOCs mostly induced distinct dynamic changes in the spike trimers affecting their local HDX (Supplementary Figs. 9–12). However, we identified common effects in all spike variants that could not be explained by individual mutations, but by an entire set of alternant substitutions and deletions.

First, increased HDX in regions spanning residues 92–110 (ß5-ß6 loop, ß6, and ß6-ß7 loop), 136-153 (ß9 and ß9-ß10 loop), 177–191 (ß10-ß11 loop) and 242–264 (ß14 and ß14-ß15 loop) of the NTD was seen in every spike variant (Fig. 2b and Supplementary Figs. 9–19). Those effects are marked and cannot be associated with the enhanced open conformation, as they were not highlighted when comparing the HDX profiles of Wuhan and G614 spikes, while they were also present in omicron spike. (Fig. 2b). This suggests that the conformational evolution of alpha, beta, delta and omicron spike variants converged into a common conformational landscape characterized by a marked increase in flexibility of the peripheral NTD sites of all VOCs.

Second, the core S2 helices of the HR1 of all spikes of VOCs showed higher HDX than those of G614 spike (Fig. 2 and Supplementary Figs. 9–12). Furthermore, the HR1 of omicron spike, which carries a unique set of mutations (Q954H, N969K, and L981F), appeared the most dynamic among the spike variants, showing a drastic increase in HDX transmitted up to the CH region (residues 945–1006) (Supplementary Fig. 12). As the HR1 and CH are parts of the spike fusion machinery[15], their enhanced dynamics suggests that all VOC spikes, and omicron spike in particular, are already pre-primed for the fusion event, even in the absence of ACE2 when compared to Wuhan and G614 spike.

## Early VOC spikes, alpha, and beta, show the highest avidity towards ACE2

Next, by studying the HDX of the ACE2 ectodomain alone and in complex with spike trimers (1:2 spike trimer: ACE2) and the isolated ancestral RBD (3:2 RBD: ACE2), we measured the magnitude of the HDX effects (ΔHDX) induced by spike binding to ACE2. The whole population of the isolated RBD is binding competent, granting complete occupancy of the ACE2 binding sites, whereas only a fraction of the RBDs embedded within spikes are erect and thus able to engage the receptor. The observed ΔHDX results from a cumulative effect of binding stoichiometry (how many ACE2 are bound) and the stability of the hydrogen-bonding network between spikes and ACE2 (which can be related to the spike-receptor binding affinity), enabling us to rank the spike-receptor binding avidity (i.e., the overall strength of binding arising from the affinity of an individual RBD-ACE2 interaction and the stoichiometry of each spike trimer engaging between zero and three ACE2 molecules at once). We followed the HDX of 167 peptides, spanning 81% of ACE2 sequence (Supplementary Fig. 20).

We observed a decrease in HDX in the ACE2 regions spanning residues in contact with the spike RBD[14,52,53] (segments 23–45, 327–356, and 80–83) and in the ACE2 segment 58–72 with G614, alpha, beta and delta spikes (Fig. 3a and Supplementary Figs. 21–27). The cumulative ΔHDX induced by different spikes varied with alpha > beta > delta > G614 > Wuhan ≈ omicron (Fig. 3b and Supplementary Fig. 28). These ΔHDX values were generally lower than that induced by the isolated RBD, indicating that a fraction of ACE2 molecules remain unbound in the spike:ACE2 states, thus suggesting that all binding-competent RBDs within the trimers are presumably fully occupied. Specifically, the ACE2 binding to G614 spike measured by ΔHDX is 3 times higher than that of Wuhan spike (and 2.5 times lower than that of the isolated ancestral RBD). As D/G614 does not directly engage in the interaction with ACE2 and previous studies employing surface-based affinity assays with immobilized spikes reported the affinities of these two spikes to be similar[26], we interpreted this difference as being solely due to an increase in binding stoichiometry induced by the higher preference of G614 spike for the open state. Alpha, beta, and delta spikes appeared to show even higher binding to ACE2, with respective 2.7-, 2.4-, and 2.2-fold increase in ΔHDX compared to G614. In their case, we ascribe it to an increase in both binding stoichiometry, which correlates with the observed higher abundance of the open state in these spikes, and higher affinity towards ACE2 reported in previous studies[26,29,54]. In contrast, omicron spike displayed a ΔHDX much lower than G614 spike and overall comparable to Wuhan spike. This is in line with our observation that omicron spike assumes predominantly a closed state, while indicating that it exhibits no or only minor increase in affinity induced by the substitutions in its RBD.

## ACE2 binding stabilizes the open state and shows signs of cooperativity

Next, we observed the HDX of spike trimers and of the isolated ancestral RBD in complex with ACE2 to gain further insight in their binding mechanism and to understand how mutations in spike variants influence this interaction. Our HDX analysis of the complexes spanned on average 77% of spike protein sequence (Supplementary Figs. 29–33).

First, the receptor-bound spikes showed marked protection towards HDX in the RBD regions 442–506 and 400–427 with varied magnitude and extension of the effect in different variants (Fig. 4 and Supplementary Figs. 34–40). This is due to the direct engagement of these regions by ACE2 and is consistent with previous structural studies that described the spike-ACE2 interface as being constituted by the RBM (residues 438–506) and the residue 417[52].

Second, we observed decreased HDX in the RBD-s, which we associated with increased opening, manifesting in all receptor-bound spikes, with this effect being the most pronounced for Wuhan spike (residues 569–582) and omicron spike (residues 552–567 and 575–583) (Fig. 4 and Supplementary Figs. 34, 39). This marked stabilization of the open conformation upon association with the receptor aligns with the preference for closed conformations that characterizes these two spikes. In contrast, the other variant spikes (G614, alpha, beta, and delta), which favour the open conformation even in the absence of

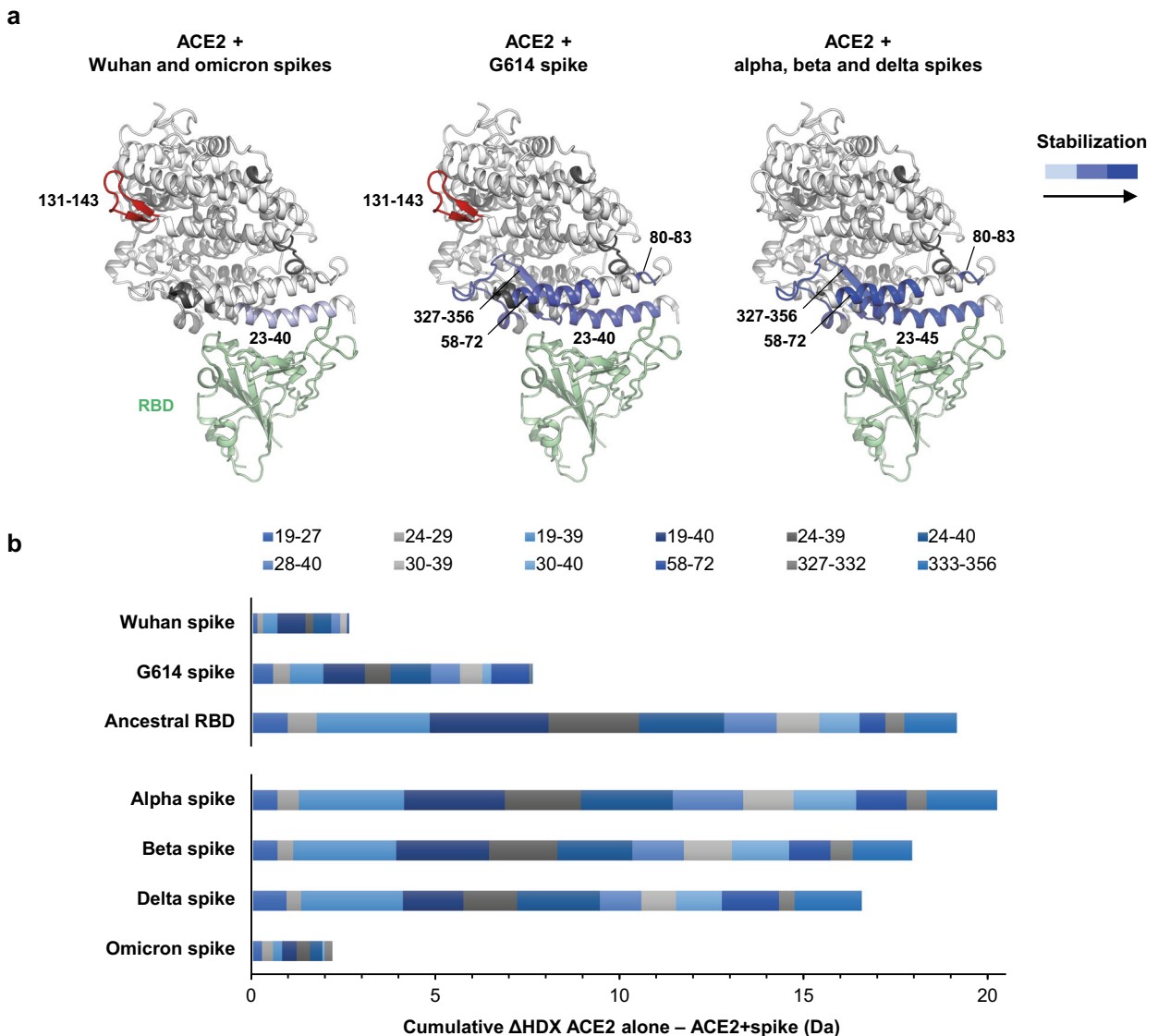

**Fig. 3 | Effect of spike binding on ACE2 dynamics. a** Regions of ACE2 manifesting a significant decrease in HDX upon spike binding are superimposed on the structure of ACE2 ectodomain bound to RBD (PDB: 2ajf), coloured in blue scale according to the magnitude of the HDX effect. The region coloured in red indicates increased HDX upon binding, in dark grey regions with no coverage. **b** ACE2 binding avidity. The cumulative difference in HDX (ΔHDX) between ACE2 alone and ACE2 bound to spikes and to the isolated ancestral RBD for selected peptides spanning binding sites and across time points 20 s on ice, 10 min at 23 °C and 360 min at 28 °C is plotted. A plot for all time points in Supplementary Fig. 28.

ACE2, showed only minor decrease in the HDX of the RBD-s upon ACE2 binding (Fig. 4 and Supplementary Figs. 35–38). Taken together, these observations confirm the idea that the RBD-s plays a significant role in the spike transition from closed to open state and indicate its further rigidification upon receptor binding.

Furthermore, we observed that the HDX profiles of all peptides spanning the RBM of spike trimers (residues 495-503) in the ACE2-bound states showed bimodal isotopic distributions, hence a high- and a low-mass population, whereas a single unimodal distribution characterized the apo states (Fig. 5 and Supplementary Figs. 41–45). These data suggest that the RBD of bound spikes can explore two distinct and slowly interconverting populations, which exchange giving rise to two isotopic distributions. The less exchanged (low-mass) population accounts for open protomers with a bound RBD. The more exchanged (high-mass) population represents RBDs perturbed by the presence of ACE2 but likely unbound: either because ACE2 transiently dissociates from them over the course of the exchange reaction or because they assume an intermediate, not fully

erect state. The former hypothesis is supported by the observation that no evidence of a population consistent with the HDX of the unbound RBMs appeared in the spike bound states at the early time points. However, the ACE2-bound state of the isolated ancestral RBD (3:2 RBD: ACE2), containing a significant fraction (33%) of unbound population, did not display bimodal isotopic distributions in the RBM under the conditions studied (Supplementary Fig. 41), suggesting that a simple mixture of bound and unbound populations of RBDs, even in the context of a spike trimer, would not manifest with an HDX bimodality either. Hence, we associated the high-mass populations of the bimodal HDX profiles with RBDs in an intermediate state, between closed and fully erect, receptor-binding-competent conformations. Such populations, characterized by disordered RBDs, have been observed and described before in cryo-EM studies[11,15,55]. We thus rationalize, also based on the spike receptor binding mode reported in previous studies[14,56], that this population reports on the trimer capability to erect additional RBDs upon ACE2 binding to the neighbouring one/-s: a sign of cooperative opening. The exact

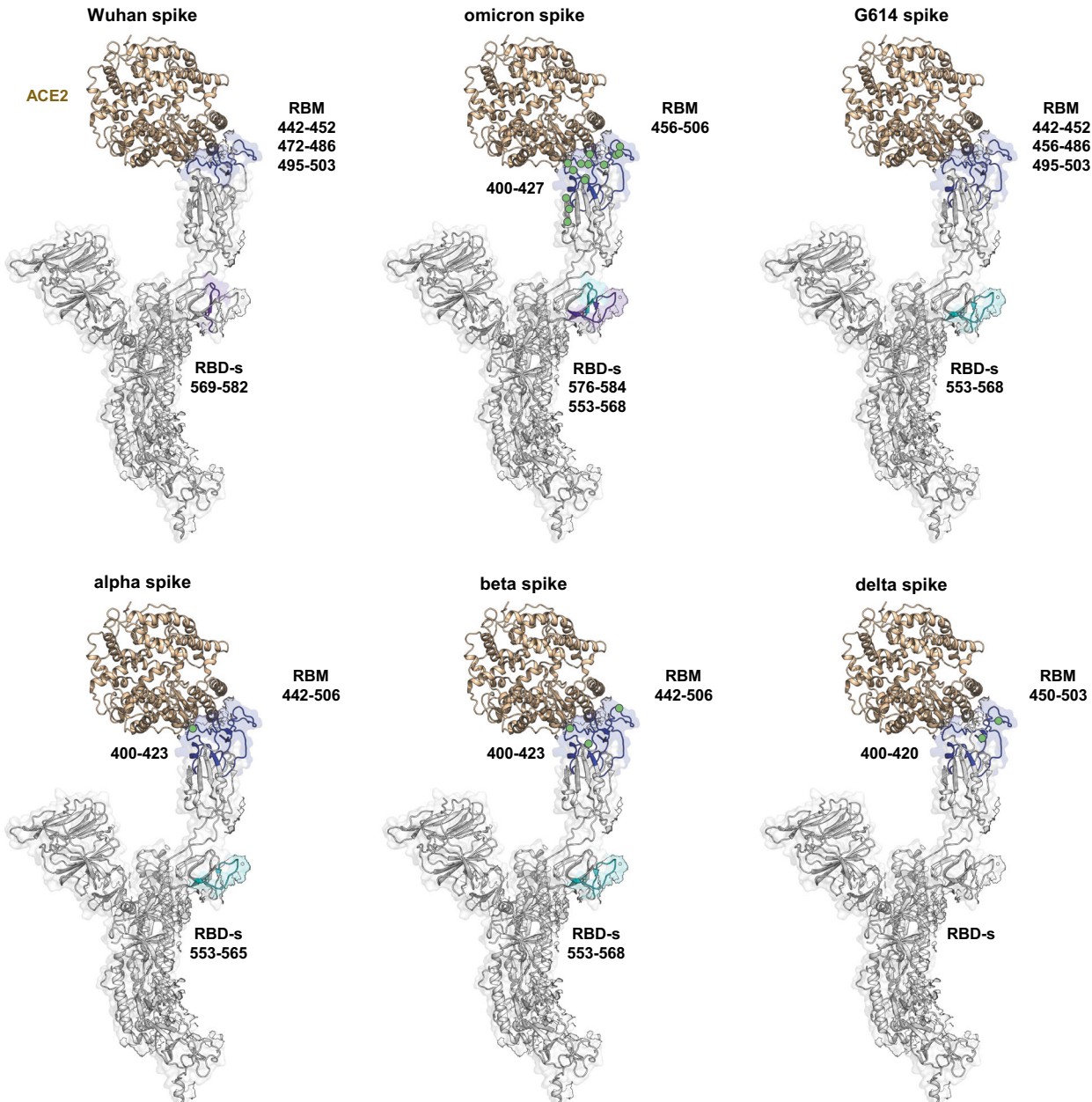

**Fig. 4 | Effect of ACE2 binding on spikes dynamics.** Regions manifesting decreased HDX in spikes when in complex with ACE2 are superimposed on one single protomer of spike D614 with one RBD bound (PDB: 7a95[14]). In blue: decreased HDX for direct binding in the RBD; in purple: marked decreased HDX in the RBD-s; in cyan: minor decrease in HDX in the RBD-s; sites of amino acid changes in the RBD are indicated with green spheres.

relative proportion of the low- and high-mass subpopulations cannot be derived from our data as we cannot accurately deconvolve the two isotopic distributions. Nevertheless, the apparent abundance of the high-mass population seems to correlate with the overall preference of a given spike to adopt the open conformation as described above. As such, G614 and delta spike displayed more intense high-mass populations compared to Wuhan spike and thus higher cooperativity of ACE2 binding, whereas beta spike displayed more intense high-mass populations compared to alpha spike. In contrast, omicron spike, which we believe assumes a predominantly closed conformation, exhibited a minimally represented high-mass population, indicating low level of ACE2 binding cooperativity (Fig. 5). From these data, we conclude that the binding energy exerted onto the erect RBD upon the association to the receptor is thus in part directed towards other still-down RBDs, leading to positive cooperative opening of the trimer upon ACE2 interaction.

## ACE2 engagement mobilizes the S2 subunit, priming spike for fusion

ACE2 receptor binding initiates a cascade of dynamic events that prime spike for fusion. By measuring the HDX increase in spike upon ACE2 engagement, we analyzed how the ACE2 binding destabilizes the spike structure and inferred how easily spike trimers of different variants are primed for fusion.

We observed that, upon ACE2 binding, the 630 loop and FPPR segments of all spikes present the same HDX changes we observed before for the closed-to-open transition, suggesting displacement of these two elements from their unbound trimer position (Supplementary Figs. 34–39). Moreover, upon ACE2-binding, all spikes manifested increased dynamics, as evidenced by increased HDX, in various protein domains, with common effects to all spikes seen in the top parts of the NTD-s and RBD-s (residues 304–387), in the part of the RBD not directly engaged in ACE2 binding (516–534), at the level of the furin-

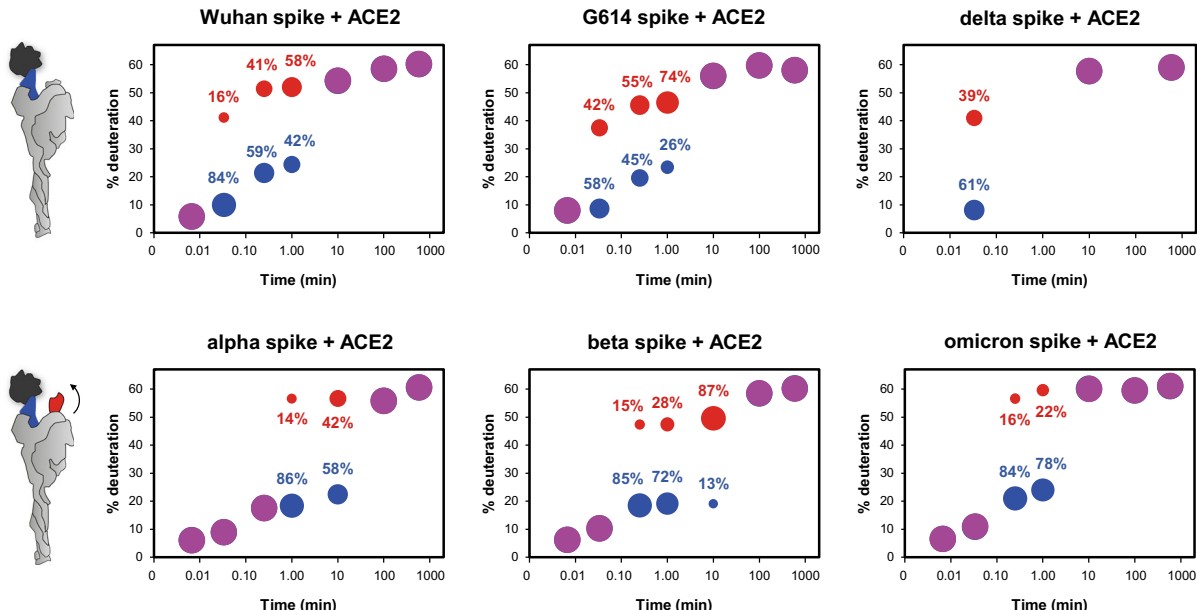

**Fig. 5 | Cooperative binding (bimodal HDX in the RBM of spikes in complex with ACE2).** The bubble plots of model peptides of the RBM provide an estimation of the level of deuteration and the relative intensity of the two spike subpopulations upon receptor binding. Data are indicative and semi-quantitative and need to be regarded as trends. The bubbles of the low- and high-mass population (bimodal distributions) are sized according to their approximated relative intensity and coloured in blue and red respectively. Magenta bubbles indicate unimodal isotopic distributions, meaning that the two populations converge, uptaking the same level of deuterium. Peptides considered: YGFQPTNGVGYQPYRVVVL (495–513) of Wuhan spike; YGFQPTNGVGYQPYRVVVL (495–513) of G614 spike; YGFQPTNGVGYQ-PYRVVVL (493–511) of delta spike; YGFQPTYGVGYQPYRVVVL (492–510) of alpha spike; YGFQPTYGVGYQPYRVVVL (492–510) of beta spike; YSFRPTYGVGHQPYRVVV (492–509) of omicron spike.

cleavage site (between 672–703), in the FP (between 764–790), HR1 and CH (923–1001); while omicron and alpha spike additionally showed increased HDX in the FP-HR1 segments 875-922 (Fig. 6a and Supplementary Figs. 34–39). Each spike trimer manifested a different sensitivity to ACE2, yet the most pronounced change in HDX upon ACE2 binding was observed in omicron spike, where the whole HR1 showed the highest increase in HDX among all spike variants (Fig. 6b and Supplementary Fig. 39). Notably, peptides spanning residues 962–982 manifested increased HDX in the form of bimodal isotopic envelopes when spikes are engaged to ACE2 (Fig. 6c and Supplementary Fig. 46). This segment encompasses the HR1 helix, which undergoes a large conformational rearrangement upon the transition to post-fusion state[15], suggesting that our analysis captured the specific dynamic events leading to the HR1 reorientation, which primes spikes for fusion. The breadth of this HDX bimodality varies with omicron > alpha > beta > G614 ≈ Wuhan spike, with omicron spike pre-manifesting it also in the absence of ACE2 (Fig. 6c and Supplementary Fig. 46).

These results also suggest that the energy gain from ACE2 binding is not only exploited to erect still-down RBDs (cooperative opening), but mainly distributed along the S2 core helices to prime spikes for the transition to the post-fusion conformation. Of note, the magnitude of destabilization (HDX increase) observed in the HR1 upon ACE2 binding inversely correlated to the extent of the cooperative opening of the various spike trimers. For instance, G614 spike bound ACE2 with higher ΔHDX than Wuhan spike (Fig. 3b and Supplementary Fig. 28), yet the observed increase in the HDX of HR1 was similar in G614 and Wuhan spike (Fig. 6 and Supplementary Figs. 34, 35), likely because the surplus of G614 spike binding energy is directed to cooperative opening of the trimer monomers (Fig. 5). Similarly, beta and alpha spikes bound ACE2 with similar ΔHDX (Fig. 3b and Supplementary Fig. 28), although the destabilization in the HR1 observed for alpha spike was higher than for beta spike (Fig. 6 and Supplementary Figs. 36, 37) because beta spike likely directs more binding energy to the cooperative effects (Fig. 5). Finally, omicron spike engaged ACE2 with ΔHDX similar to Wuhan

spike (Fig. 3b and Supplementary Fig. 26), although its HR1 became highly destabilized upon binding, much more than in Wuhan spike (Fig. 6 and Supplementary Figs. 34, 39). As omicron spike manifested lowly intense high-mass populations in the RBM, which were much less intense than those in Wuhan spike (Fig. 5), it seems plausible that omicron spike directs the majority of the binding energy to the trimer core helices, rather than promoting a cooperative effect to enhance further opening and receptor binding. Also, given that the core helices of omicron spike appeared already highly dynamic even in the absence of ACE2 (Supplementary Figs. 12, 46), we propose that omicron evolved structural features that make it more 'ready' for the conformational rearrangements associated with fusion, whereas spikes of previously circulating variants evolved to promote opening and stronger receptor engagement.

## Discussion

Our HDX-MS analysis of spikes of SARS-CoV-2 VOCs reveals a number of adaptations that improved their ability to infect humans. First, we show that, although spikes of different VOCs harbour very different sets of mutations in the NTD, the effect that these exert at the protein conformational level converges, resulting for all of them in a significantly enhanced flexibility of the peripheral portion of the NTD (Fig. 2b). Previous studies indicate that this part of the NTD, made of mobile loops and ß-hairpins, is stabilized by a network of hydrophobic and electrostatic interactions in Wuhan spike[11,57], which we now demonstrate to be disrupted in spikes of VOCs. As this divergence hotspot, identified here, strongly overlaps with the NTD antigenic supersite[58,59], we postulate that the increased flexibility acquired by the alpha, beta, delta, and omicron variants enables their escape of NTD-targeting antibodies[27,32,60–65], endowing these strains with an evolutionary advantage. The function of the NTD in spike is still poorly understood, but our results suggest that it could be used as a large and easily modifiable antigenic surface, allowing the virus to divert the host immune responses to it rather than to the RBD, which has the crucial role in the viral cycle. The conformational plasticity of this NTD site

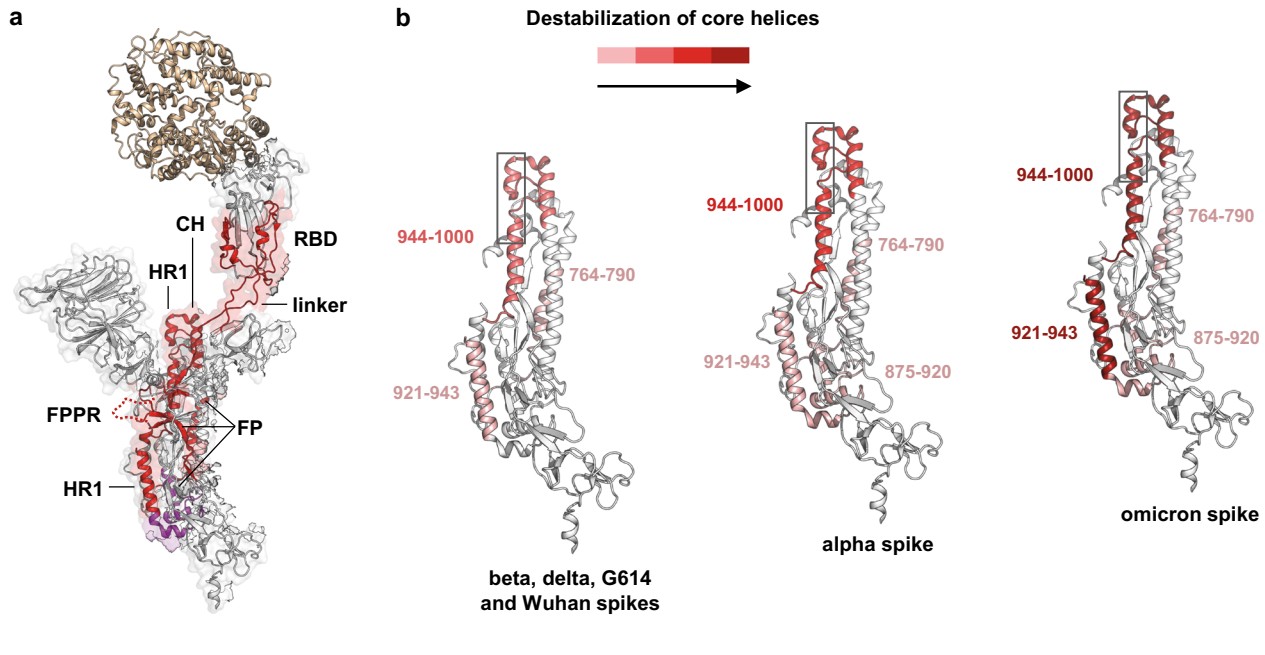

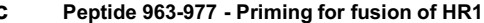

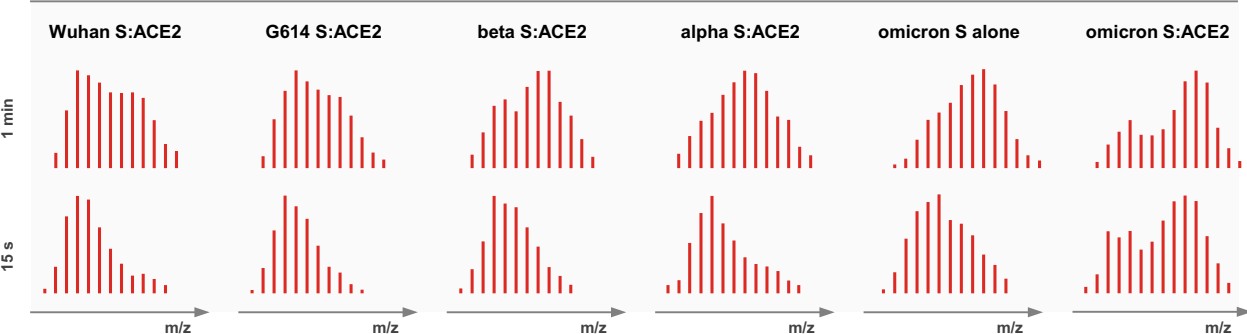

**Fig. 6 | Spike priming for fusion. a** Regions manifesting increased HDX in spikes bound to ACE2 are superimposed and coloured in red on a single protomer of the structure of D614 spike with one RBD bound (PDB: 7a95[14]); regions with increased HDX only in spike of alpha and omicron variants bound to ACE2 are coloured in magenta. **b** The magnitude of the destabilization of the core helices is represented by differential colouring (red scale) for the various spike trimers. The HR1 region (962–982) manifesting HDX bimodality is framed in grey. **c** The bimodal isotopic envelopes of a model peptide of the HR1 region 962–982 are shown at 15 s and 1 min time points for spike (S) variants in the ACE2-bound state and for omicron spike alone, to exemplify the priming for fusion upon receptor engagement. Bimodal envelopes manifest in omicron spike also in the absence of ACE2, indicating it as pre-primed for fusion.

presumably allows antibody-vulnerable configurations of conformational epitopes to remain occasionally populated, reducing, but not completely abrogating, antibody-mediated protection. At the same time, linear NTD epitopes are likely more impacted by the residue changes in spike variants rather than by a different conformational aspect of the NTD of VOCs.

The capability of SARS-CoV-2 spike to transition to the open receptor-binding-competent conformation has changed throughout the virus evolution, as shown in many structural studies[26]. The acquisition of D614G substitution by Wuhan spike was the turning point, as it enhanced trimer opening[18–21] and virus transmissibility[22–24], promoting the emergence of all VOCs. Comparing the HDX of G614 spike to Wuhan spike allowed us to identify dynamic events underpinning the closed-to-open state transitions. Our data reveal an opening mechanism where the RBD-s (SD1) subdomain functions as a lever that modulates the opening of the trimer, by coordinating an allosteric crosstalk between the RBD, D/G614 in the NTD-s (SD2), and the FPPR of a neighbouring monomer. In summary, upon disruption of the D614-K854 salt bridge, we show that the FPPR becomes more dynamic. The FPPR abuts the RBD-s, which abuts the RBD. As an ordered FPRR would clash against the RBD-s of a protomer in the open conformation[15], in

G614 spike, a more flexible FPPR better allows the RBD-s to rigidify and adopt the optimal conformation enabling the RBD to erect. We found this allosteric axis being bidirectional, as those elements also change configuration when the open conformation is stabilized upon receptor binding to the RBD. This allosteric axis is conserved in early VOCs, as their spikes share G614 and favour the open state, but is no longer retained in omicron spike. We rather observed that omicron spike is preferentially closed, which is in line with some previous cryo-EM studies, especially those which employed spike constructs lacking pre-fusion stabilizing mutations[34,66,67]. Most cryo-EM analyses of 2P omicron spikes (the construct used in this study) reported preponderance of open states when compared to Wuhan spike (but not necessarily to G614 spike);[68–71] however, specific conformations and conclusions reported by each of these studies varied significantly. Here, we suggest that the omicron substitution N856K, which changes the network of interactions between the S2 and D568 and T572 in the RBD-s[32], is responsible for restoring the closed-trimer position of the RBD-s, hindering the RBD/RBD-s/FPPR crosstalk. Furthermore, by observing the HDX of omicron spike, we infer that the structural organization of the 630 loop in the NTD-s is also dependent on the position of the RBD-s. These observations are in line with the recent structural studies on

omicron spikes without pre-fusion stabilizing mutations[66,67] and prompt us to believe that the RBD-s plays the central role in the spike opening mechanism. In line with this, a very recent report has demonstrated that a monoclonal antibody recognizing an epitope in the RBD-s induces spike to open and is able to neutralize alpha, beta, delta and omicron VOCs[72], thus making the RBD-s also a promising target for therapeutic antibodies against SARS-CoV-2.

When in open conformation, spike binds to ACE2, which primes it for the subsequent fusion event[14]. The transition into the postfusion state involves dramatic rearrangements of the trimer structure, with the S1 subunit dissociating and the HR1 and CH subdomains of S2, which in prefusion state comprise short separate antiparallel helices, reorienting to form a long continuous stem helix[15,73,74]. Our HDX studies on prefusion-stabilized spikes allowed us to better understand the dynamics and energetics of the ACE2 association and the early effects of the binding event on the spike trimers, before full transition to the postfusion state. We reveal that the ACE2 binding energy is distributed between both the S1 and S2 subunits. The ACE2 binding to one S1 subunit promotes additional protomers to open in a cooperative manner, causes rigidification of the RBD-s and the expulsion of the 630 loop from its unbound trimer position, which likely exposes the abutting S1–S2 cleavage site for easier protease cleavage. In the S2 subunit, the FPPR becomes more dynamic likely enabling the fusion peptide to engage the host cell membrane while also exposing the proximal S2′ site for cleavage. The 630 loop and the FPPR are therefore not only involved in the transition to open conformation, but also appear directly involved in the events leading to the fusogenic transition. The S2 core helices of HR1 and CH become highly destabilized upon receptor binding, readying for the subsequent major conformational changes. We demonstrate that the distribution of binding energy between the S1 and S2 subunits varies depending on the VOC. The alpha, beta, and delta variants favour open conformations of the spike and bind ACE2 with high avidity, employing high binding energy to open more protomers and to prime the trimer for fusion. Conversely, the omicron spike is preferentially closed and shows similar avidity towards ACE2 to Wuhan spike, yet it directs most of its binding energy towards S2, without further protomer opening in the S1, thus optimizing its readiness for fusion. This, combined with the unique set of substitutions in the HR1 of omicron spike that make its S2 core helices highly dynamic even in the absence of ACE2, suggests that omicron evolved a different mechanism to utilize its spike compared to other VOCs, optimizing its fusion rather than the adhesion function, which enables it to be better primed for fusion than other spike variants and with minimal receptor binding.

Our observations suggest that the evolutionary trajectory of SARS-CoV-2 has therefore recently deviated. The strains previously circulating had likely been selected because their open spikes facilitated receptor binding, while antibodies targeting erect RBDs were still rare in the in the context of population immunity. In contrast, we propose that the omicron variant has been selected because its closed spike hides the RBD epitopes now recognized by those antibodies, elicited by vaccination and the previously circulating alpha, beta, and delta variants, while retaining infectivity, likely due to a lesser energetic cost of initiating the fusion reaction (Fig. 7). However, the conformational evolution of spikes of VOCs displays convergence at the level of the NTD, which shows a conserved conformational aspect through the virus evolution.

Based on the last observation, it is reasonable to speculate that a variant-based vaccine could elicit antibodies adapted to the currently circulating and, potentially, to emerging VOCs as well. Recently, a COVID-19 booster candidate modelled on beta spike was shown to induce higher neutralizing antibody titres against multiple SARS-CoV-2 variants, including omicron, than the Wuhan spike-based boost[75]. In line with this, we observed that beta spike, in addition to its VOC-NTD features, displays conformational features analogous to both alpha

and omicron spikes at the level of the RBD, whilst its higher degree of opening compared to all other VOC spikes presumably enables efficient S1 epitope exposure.

Our HDX-MS analysis thus provides insights into the structural biology and dynamics of spikes of VOCs, supporting the research of the next-generation SARS-CoV-2 vaccines.

## Methods

### Protein production
The constructs coding for Wuhan spike and variants G614, alpha, beta, delta, and omicron were based on the furin-uncleavable version of the SARS-CoV-2 spike protein ectodomain (residues 1–1208) with a set of stabilizing mutations (R682S, R685S, K986P, and K987P) and a foldon trimerization tag, described before[11]. Spikes were all produced as described before[26]. In addition to the stabilizing mutations, variant spikes had the sets of mutations as specified in Supplementary Fig. 1 (except substitutions at position 681 directly next to the furin-cleavage in alpha and delta spikes, as this site has been made inactive anyway by the stabilizing mutations).

Briefly, they were expressed in Expi293F cells (Gibco) growing in suspension at 37 °C in an 8% $CO_2$ atmosphere transfected with ExpiFect-amine 293 (Gibco) and 1 mg of DNA per litre of culture. The enhancers were added 20 h after transfection and the cells were then moved to 32 °C. The supernatant containing the protein was harvested on the fifth day post-transfection. The collected supernatant was then clarified, bound overnight to TALON beads (Takara), briefly washed, and eluted with 200 mM imidazole. The ACE2 ectodomain (residues 19–615) and the isolated ancestral RBD (residues 319–514) were made exactly as described before[11,76]: expressed in Expi293F cells and purified with Strep-Tactin XT resin. Following affinity chromatography, all proteins were concentrated and either flash-frozen or gel-filtered on a Superdex 200 Increase 10/300 GL column (GE Life Sciences) into a buffer containing 20 mM Tris and 150 mM NaCl (pH 8.0).

### Peptide identification
Prior to conducting HDX-MS experiments, peptides were identified by digesting undeuterated spikes and ACE2 using the same protocol and identical liquid chromatographic (LC) gradient as detailed below and performing MS$^E$ analysis with a Synapt G2-Si mass spectrometer (Waters), applying collision energy ramping from 20 to 30 kV. Sodium iodide was used for calibration and Leucine Enkephalin was applied for mass accuracy correction. MS$^E$ runs were analyzed with ProteinLynx Global Server (PLGS) 3.0 (Waters) and peptides identified in 3 out of 4 or 2 out of 3 runs, with at least 0.2 fragments per amino acid (at least 2 fragments in total) and mass error below 10 ppm were selected in DynamX 3.0 (Waters). The MS traces of peptides were further visually inspected to exclude those of insufficient quality or false identifications. False identifications were also discarded by cross-matching peptides containing residue changes among spike variants. To note, post-quenching de-glycosylation was not performed and no attempt to identify glycosylated peptides was made, causing partial sequence coverage in proximity to glycan sites. Coverage in 6 out 23 glycosylation sites (N282, T323, S325, N1158, N1173, N1194) arose from non-glycosylated peptides of spike molecules of incomplete glycan occupancy.

### Optimization of the HDX conditions
Prior to performing the deuterium labelling under several time points and temperatures, the spike: ACE2 binding stoichiometry most suitable for HDX-MS experiments was optimized. Wuhan and G614 spikes alone and incubated with ACE2 at ratio 1:2 and 1:3 spike: ACE2 were labelled for 20 s on ice. ACE2 alone and incubated with Wuhan spike at ratio 1:2 and 1:3 spike: ACE2 was labelled for 100 min. The exchange reaction and the LC-MS analysis were conducted as detailed below. The binding stoichiometry

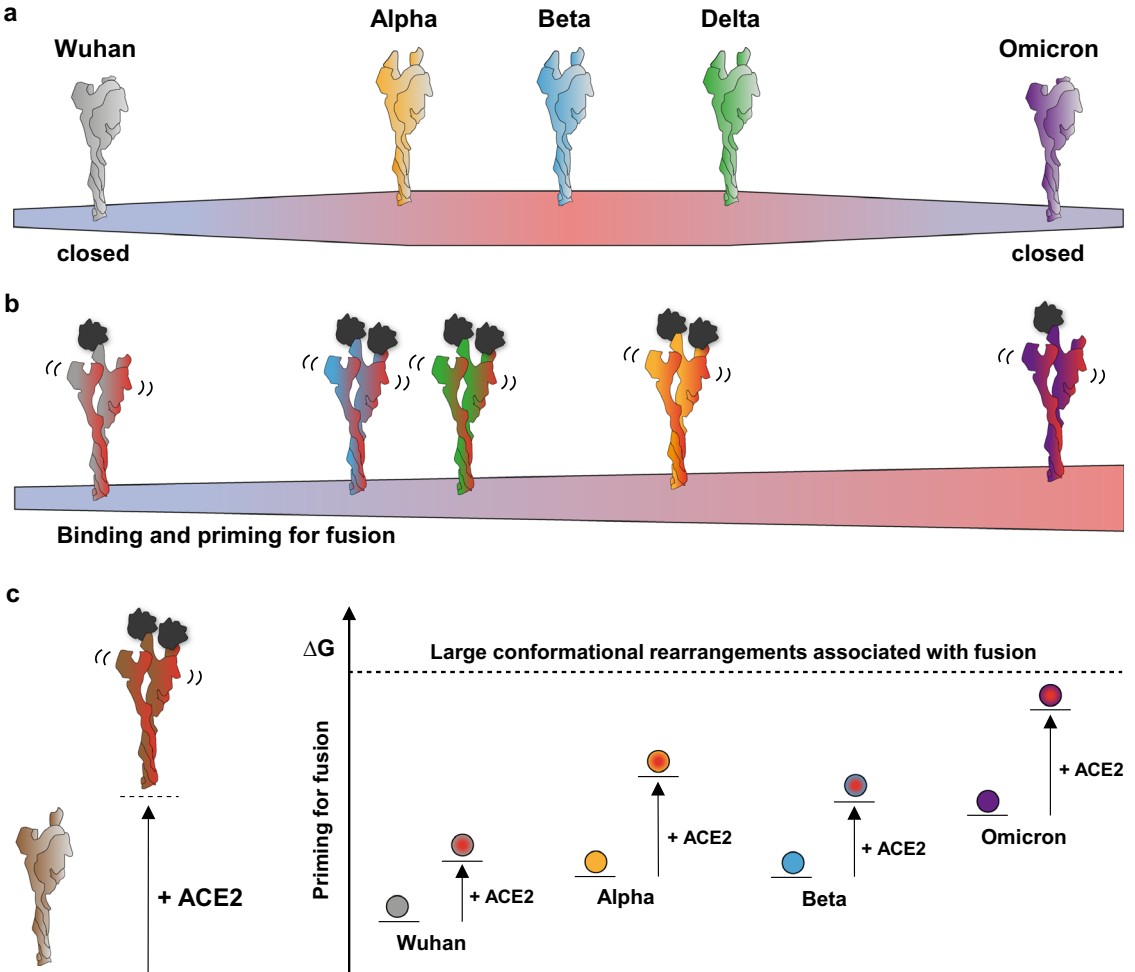

**Fig. 7 | Schematic of the structural transition of prefusion spikes. a** Preference for the open state of spikes: Wuhan and omicron spikes favour the closed state, whereas alpha, beta and delta spikes favour the open state. **b** Priming of spikes upon binding to the ACE2 receptor: alpha, beta and delta spikes bind ACE2 with higher avidity and prime for fusion better than Wuhan spike; omicron spike is primed for fusion better than the other variant spikes and with minimal receptor binding. **c** Energy landscape of spike in prefusion state. Spikes of alpha and beta variants have more dynamic core helices (higher internal energy) than G614 spike, enhancing their readiness for fusion. Omicron spike presents the most dynamic core helices among all spike variants, which highly increases its internal energy and makes it pre-primed for fusion in the absence of ACE2.

enabling to maximize the magnitude of HDX binding effects for both proteins was identified as 1:2 spike: ACE2 (Supplementary Figs. 47, 48). For deuterium labelling, we used freshly prepared spike protein batches, as HDX-MS data collected in-house on previously frozen samples showed the presence in the peptide MS trace of bimodal isotopic distributions, likely representing two predominant populations (a folded and an open-trimer population). These were seen for peptides of the same domains highlighted by Costello et al.[36]. We could relate the relative intensity of the two populations to the freezer storage time. As the biological relevance of the open-trimer conformation adopted by recombinant spike and whether it is sampled by the native membrane-bound trimer remain to be elucidated, we conducted our experiments on freshly produced spike batches, where we instead observed unimodal isotopic envelopes, representing a single population of folded spike.

**Hydrogen-deuterium exchange (HDX)**
To expand the dynamic and time window studied, the HDX was conducted at three different temperatures (on ice, at 23 °C and 28 °C in a thermomixer). Before initiating the exchange reactions, spikes and ACE2 were incubated alone or in complex at ratio 1:2 spike trimer: ACE2 for one hour at the selected labelling temperatures (i.e., on ice or

in the thermomixer at 23 °C or 28 °C) and the labelling buffer was as well temperature equilibrated. After equilibration, the deuterium labelling was conducted for 4 s and 20 s on ice; 15 s, 1 min, 10 min and 100 min at 23 °C; 6 h at 28 °C. The exchange times on ice (~0 °C) can be converted into approximately 0.36 and 1.78 s at 23 °C and the exchange time at 28 °C can be converted into approximately 585 min (~10 h) at 23 °C[77]. The exchange reactions were initiated by 6.25-fold dilution into deuterated PBS buffer (pH$_{read}$ 7.35), yielding to 84% of final deuterium content in the reaction mixture, and quenched by 1:1 dilution into ice-cold 100 mM phosphate buffer containing 4 M Urea and 0.5 M TCEP (final pH$_{read}$ 2.3). Samples were held on ice for 30 s and snap-frozen in liquid nitrogen. Triplicates were performed for time points 20 s on ice and 10 min at 23 °C; duplicates were performed for time point 4 s on ice. For delta spike, the HDX at fewer time points was sampled. In the apo state, we performed 4 s (duplicates) and 20 s on ice (duplicates); 10 min at 23 °C (duplicates) and 360 min at 28 °C (duplicates). For the ACE2-delta spike, we performed 20 s on ice (triplicates); 10 min at 23 °C (singlet) and 360 min at 28 °C (singlet). For omicron spike, the time point of 20 s on ice (both states) was measured in duplicates instead of triplicates. Maximally labelled samples were produced by labelling spikes and ACE2 in 6 M deuterated Urea in D$_2$O and 20 mM TCEP, resulting in a final deuterium content as for the other labelled samples. The maximally labelled controls (MaxD) were

quenched after 6 h by 1:1 dilution with ice-cold 100 mM phosphate buffer (final $pH_{read}$ 2.3), held for 30 s on ice and snap-frozen in liquid nitrogen. The isolated ancestral RBD of Wuhan spike and ACE2 were incubated alone and together at ratio 3:2 RBD: ACE2, simulating the binding stoichiometry of spikes: ACE2 complexes. After equilibration, deuterium labelling was performed with a 6.25-fold dilution into deuterated PBS buffer and the reactions quenched after 20 s on ice (triplicates), 10 min at 23 °C (triplicates) and 360 min at 28 °C (duplicates). Samples were held on ice for 30 s and snap-frozen in liquid nitrogen. The labelled samples were kept at −80 °C until LC-MS analysis.

### Protein digestion and LC-MS

Frozen protein samples were quickly thawed and injected into an Acquity UPLC M-Class System with HDX Technology (Waters). Proteins were on-line digested at 20 °C into a dual protease column (Pepsin-Type XIII 1:1, NovaBioAssays) and trapped/desalted with solvent A (0.23% formic acid in MilliQ water, pH 2.5) for 3 min at 200 μL/min and at 0 °C through an Acquity BEH C18 VanGuard pre-column (1.7 μm, 2.1 mm × 5 mm, Waters). Peptides were eluted into an Acquity UPLC BEH C18 analytical column (1.7 μm, 2.1 mm × 50 mm, Waters) with a linear gradient raising from 8 to 40% of solvent B (0.23% formic acid in acetonitrile) at a flow rate of 40 μL/min and at 0 °C. Then, peptides went through electrospray ionization in positive mode and underwent MS analysis with ion mobility separation. To eliminate peptide carry-over, the protease column was washed between each run of deuterated samples using 1.5 M Gu-HCl in 100 mM phosphate buffer (pH 2.5) and a run with a saw-tooth gradient was carried out for washing the chromatographic segments.

### HDX-MS data analysis

HDX bimodal envelope distributions were analyzed by HX-Express2[78] after spectra smoothing 8 × 4 Savitsky-Golay in MassLynx. The deuterium uptake (DU) at the peptide level was calculated with DynamX 3.0 and the statistical analysis was performed with Deuteros software[79], applying 98% or 99% confidence interval. To compare peptides containing residue substitutions in spike variants (mutant peptides) with peptides of G614 spike, segments with identical N- and C-termini were selected. Their difference in deuterium incorporation (ΔHDX) was calculated according to Eq. 1 and plotted in Supplementary Figs. 9–12:

$$\triangle HDX = \left( \frac{DU \text{ mutant peptide}}{DU \text{ MaxD mutant peptide}} \times DU \text{ MaxD G614 spike peptide} \right) - DU \text{ G614 spike peptide}$$

(1)

To estimate the impact of the difference in chemical exchange rate constants ($k_{ch}$) on the observed ΔHDX between mutant peptides and peptides of G614 spike[50], firstly the $k_{ch}$ of individual residues within the spike protein sequences were calculated[49]. Successively, at peptide level, the percentage of difference in $k_{ch}$ (%$\Delta k_{ch}$) were compared to the percentage of ΔHDX normalized by the MaxD (%ΔHDX) in the time point showing maximal effect, as reported in Supplementary Data 1. The identified differences in HDX between spike variants and G614 spike in segments spanning amino acid changes resulted of high confidence, with the impact of $k_{ch}$ negligible, except for peptide 946–961 of delta spike.

### Reporting summary

Further information on research design is available in the Nature Portfolio Reporting Summary linked to this article.

## Data availability

According to the community-based recommendations[80], to allow access to the HDX data of this study, the HDX summary tables are included as Supplementary Tables 1–19, the HDX data tables are provided as source data with this paper and all deuterium uptake plots are deposited in the figshare repository at https://figshare.com/s/f1db735795acd604e470. The HDX mass spectrometry data have been deposited to the ProteomeXchange Consortium via the PRIDE partner repository with the dataset identifiers PXD039760 and PXD039762. The protein structures from other publications referenced in this paper are accessible under the PDB accession codes 7BNN, 6ZGE, 7BNO and 7A95. Source data are provided with this paper.

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

## Acknowledgements

A.P., V.C., and R.T.B. were supported by the Leverhulme Trust (RPG-2019-178) to A.P. A.P. was supported by an EPSRC Research Fellowship (EP/V011715/1). VC was also supported by a King's College London funded research associate position to E.R. J.T. was supported by OMass Therapeutics via an EPSRC iCASE Award to A.P (EP/R513064/1). R.B.P. was supported by the Analytical Chemistry Trust Fund via the Analytical Chemistry Summer Studentship scheme and by a King's College London Wellcome Trust Biomedical Vacation Scholarship (220280/Z/20/Z) to VC. D.H., and E.R. were supported by a UKRI Future Leaders Fellowship to E.R. (MR/S015426/1). A.G.W., D.J.B., C.R., and S.J.G. were supported by the Francis Crick Institute, which receives its core funding from Cancer Research UK (CC2060), the Medical Research Council (CC2060), and the Wellcome Trust (CC2060). M.H.M. and K.J.D. were supported by a Huo Family Foundation Award, the MRC Genotype-to-Phenotype UK National Virology Consortium (MR/W005611/1), and an EDCTP2 programme supported by the European Union (RIA2020EF-3008 COVAB).

## Author contributions

V.C., A.G.W., and A.P. designed the research. A.G.W., D.J.B., and C.R. expressed and purified the proteins. V.C. designed and performed HDX-MS experiments and analyzed HDX-MS data. J.T., D.H., R.B.P, and R.T.B. provided support with HDX-MS data analysis. V.C. and A.G.W. interpreted the results. K.J.D., E.R., M.H.M., S.J.G., and A.P. provided resources and/or supervision. S.J.G. and A.P. supervised the research. V.C., A.G.W., E.R., S.J.G., and A.P. wrote the original manuscript with inputs from all authors.

## Competing interests

The authors declare no competing interests.
