## [Peer Review File · Nature Communications]

REVIEWER COMMENTS

Reviewer #1 (Remarks to the Author):

In the manuscript titled “Structural dynamics in the evolution of 1 SARS-CoV-2 spike glycoprotein”, Calvaresi et al. utilize HDX-MS to study the structure and dynamics of the SARS-CoV-2 spike protein. They relate their observations to published structures of the spike.

Several of the results presented are consistent with expectations based on existing structural data and/or add to our understanding of the SARS-CoV-2 spike structure and dynamics. These include the increase in HDX in regions of the spike where increased flexibility/mobility is expected because of the D614G mutation, the decreased HDX in the regions spanning the ACE2-RBD interface, ACE2-induced changes in the S2 subunit, and the observation of increased inner flexibility of the Omicron spike RBD that agrees with observations of differing thermostability of the Omicron RBDs ([https://www.cell.com/cell-reports/fulltext/S2211-1247\(22\)00798-7](https://www.cell.com/cell-reports/fulltext/S2211-1247(22)00798-7)).

There are other observations, however, that do not agree with structural data despite the authors’ claims that they do. Notable among these is the observation, which is also a primary result/conclusion of this study that the “substitutions in omicron spike lead to predominantly closed conformations, presumably enabling it to escape antibodies”. The constructs studied in this paper include the 2P mutations. Published cryo-EM structures of Omicron spikes that include the 2P mutations show a preponderance of open states. Two examples of these are:

<https://www.nature.com/articles/s41467-022-28882-9>

<https://www.sciencedirect.com/science/article/pii/S2211124722001528>

The HDX results presented do agree well with structural results obtained without the 2P mutations as described in these papers (which, by the way, are not cited):

[https://www.cell.com/molecular-cell/fulltext/S1097-2765\(22\)00266-0](https://www.cell.com/molecular-cell/fulltext/S1097-2765(22)00266-0)

[https://www.cell.com/cell-reports/fulltext/S2211-1247\(22\)00798-7](https://www.cell.com/cell-reports/fulltext/S2211-1247(22)00798-7)

Indeed, the agreement is quite striking not only with the results related to the higher proportion of the closed conformation, but also the changes in the S2 region that pre-dispose the Omicron spikes to undergo structural changes required for fusion. Bottomline, the differences observed between 2P and non-2P Omicron spikes, especially related to proportions of closed vs open conformations, are now well documented, and the results presented here, agree more with what was observed in non-2P spikes, although the spikes used for the HDX-MS experiments include the 2P mutations. This is an important discrepancy that must be addressed.

Other issues:

The title of figure 1 “Mechanism of transition from closed to open states” is misleading and over-reaching. This should be rephrased to indicate what the figure is showing, ie., Differences in HDX between Wuhan D614 and G614 spikes.

Line 150: “Hence, in all analyses we used a stabilized version of spike ectodomains containing ‘2P’ mutations, which make them furin-uncleavable and unable to transition to the postfusion conformation.” The 2P mutations do not have the spike furin uncleavable, the RRAR to GSAS substitution in the SD2 subdomain does.

In Figure 2, panel A, “D614 vs G614 (closed state)” not clear what the “closed state” here indicates.

Reviewer #2 (Remarks to the Author):

This manuscript describes H/D Exchange of the intact COVID spike protein, comparing all major variants of concern for shifts in conformational dynamics associated with 'open/closed' and 'fusion priming'. The paper is well written, the data are well presented and the results provide some important insights about mutation-dependent dynamic shifts in the NTD, Ace2 binding and fusion-priming. The evidence unambiguously supports the conclusions. I have only a few minor suggestions:

1. The Authors could do a better job of citing previous HDX studies on COVID spike, some of which provide highly relevant foundation and corroborating evidence for the current work. In particular, the work of Ganesh Anand should be cited. Also, the Wilson and Komives groups have measured dynamics of Spike by HDX in various contexts that may be relevant.

2. The 'bubble plots' are an excellent way of representing EX1 kinetics. It would be interesting to see these for more than one peptide in this region, perhaps in supplemental. To have full confidence in these data, similar kinetics for overlapping or nearby peptides should be included.

3. The idea of increased dynamics as a way of evading antibody binding is an interesting one. The authors may wish to reflect on how this mechanism would impact a continuous vs. discontinuous epitope in the discussion. (At first pass, my thought would be that this mode of evasion would be much more impactful on discontinuous epitopes than on continuous ones, and that it would never offer 'complete' protection, since the 'vulnerable' configuration would still occasionally be populated... Essentially, manipulation of conformer selection for antibody binding).

Reviewer #3 (Remarks to the Author):

The manuscript by Calvaresi reports an extensive HDX-MS comparative analysis of various CoV2 spike proteins in the unbound or ACE2-complexed states. The data are well presented, and the study can reveal some useful insight into the solution behavior of the spike protein. I applaud the authors for using freshly purified proteins to minimize confounding effects of freeze thaw that have previously been observed. The methods and the majority of the analysis is solid, including the direct comparison of the solution behavior of all the variants. There are a few major concerns going back to the experimental design for the ACE2 bound complexes as outlined below, along with a other minor issues listed below that need to be addressed.

Major concern:

Why did the authors use a ratio of 1:2 spike:ACE2? Won't this mean that 1 of the 3 RBDs of the spike is unbound? For the most conclusive HDX studies it is really important to favor full binding of your protein of interest so that you measure the full extent of changes associated with binding. With the 1:2 ratio used here, only a maximum of 66% of the spike RBD population can be in the bound form. This may manifest as two apparent populations for regions that are most perturbed by ACE2 binding.

Were bimodal spectra observed across different regions of the spike beyond the 495-513 region that was presented? Additionally, how long was the spike:ACE2 complex incubated prior to exchanges? Do the authors know if this time was sufficient for equilibration to the 66% bound form?

Bimodal spectra at the 495-513 region presented look convincing. Based on the authors coverage maps each of the mutants have several peptides that span this same region in the spike proteins. The authors need to check and make sure that the bimodal (or peak broadening) is evident in all of those overlapping peptides too. I do not think it is critical for the authors to fit and thoroughly analyze all of the overlapping peptides, as it is not a central conclusion of the paper, but at the least the authors need to validate that it is observed consistently among the overlapping peptides and make a note of this in the results.

While the peak broadening is evident by eye, it also looks like the separation of the two populations within the bimodal is poor. Because of this, I do not think that there is high confidence associated with the fitting to extract the exact deuterium incorporation and exact sizes for the two populations. The authors interpret the bimodal as EX1 kinetics, but the two populations could easily result from the 66% that is ACE2 bound and the remaining 33% that is unbound. There will likely also be a distribution of different stoichiometries of ACE2-spike among the different spike molecules. For example there will be some finite number of spikes with all three lobes bound to ACE2, the majority with 2 lobes bound, a small fraction of only a single lobe bound, and a really small population of completely unbound spike. These different populations can easily result in complicated bimodal mass envelopes that have nothing to do with true EX1 kinetics. One thing the authors can do is to see if the higher deuterated population appears consistent with what was observed in the unbound spike. If bimodal profiles are observed in both the spike and the ACE2 then the authors might be able to interpret what fraction of the protein was actually bound.

These issues could have been alleviated if the experiment was designed with a sufficient excess of ACE2 to ensure fully bound complexes. I appreciate that these reagents are challenging to make and the experiments are difficult to carry out, and do not expect the authors to go back and redo the experiment. Furthermore, I think the data in this section can still be insightful, but the authors need to reexamine and rewrite this portion with full acknowledgment of all the confounding factors that limit how much can be concluded. The problem of incomplete ACE2 binding may also be a confounding factor for the following section on the S2 subunit priming.

Major point 2:

The authors examine HDX changes within ACE2 in complex with the various spike constructs to assess the binding affinity and avidity. This section is highly problematic. I am having trouble trying to understand the logic of why this experiment would reveal what the authors suggest. I agree that tighter binding will lead to more protection in the ACE2. Based on the methods the authors preincubated at a ratio of 1:2 spike trimer:ACE2. This ratio should correspond to a 3:2 ratio of spike monomer:ACE2, and if the interaction is of sufficient affinity then the vast majority of the ACE2 should be bound. For HDX studies you typically want an excess of the ligand with sufficient preincubation so you can be sure that you have near-full occupancy of the binding sites in the protein of interest. If not then you risk looking at a mixture of bound and unbound species that will depend on affinity and possibly incubation time. Depending on the association/dissociation kinetics this can also result in bimodal isotopic envelopes as the data might reflect a combination of free and bound populations, further complicating the analysis.

The authors suggest that since previous binding studies have indicate similar affinities between WT and G614, that a higher portion of the ACE2 is bound in the presence of G614 spikes. While this may be true, this would indicate that the complexes were not incubated long enough to reach equilibrium. At equilibrium with equal affinities there should be an identical amount of complex formed for both WT and G614 spikes. If there isn't then it might relate to the formation of the ACE2-WT complex being much

slower, perhaps because more time is needed for the WT to sample open conformations capable of binding.

Additionally, if the binding is incomplete then I would expect to see a bimodal isotopic profile near the binding site reflecting the population of bound and unbound ACE2. This bimodal may actually be a much more direct way to assess what portion of the ACE2 is able to bind the spike. However, in many cases the two populations in bimodal spectra are not well-resolved and quantifying the populations is difficult to determine with any confidence. Overall, this assay has a lot more variables and caveats than the authors account for, and I would be very hesitant to conclude anything from this data. I'm guessing the authors thought they could extract some additional information for the ACE2 since it's already in the dataset, but this is not a well-designed experiment for any solid conclusions. It might be a useful assay for future studies to examine kinetics of binding among the variants, but since this is not an integral part of the current paper, I advise the authors simply omit this section.

Minor point:

The authors should make a note that there is a caveat when comparing peptides among spike variants where the peptide sequences are not perfectly matched. The intrinsic rates of exchange will be offset by some of the amino acid substitutions, so it is possible to observe different exchange kinetics even if there is no actual difference in the structural dynamics of that region of the protein. The authors should make a note how they handled the comparison at regions where peptide sequences were not matched. It should help that the authors had a fully deuterated control so they can compare % deuteration instead of deuterium uptake, but this won't alleviate the caveat entirely. For example, one of the largest changes seen in with beta spikes was attributed K417N, and from the coverage map it looks like all of the peptides reporting around this region span residue 417. The authors should check how much of an effect the mutation will have on the intrinsic exchange rate. In many cases it will be relatively minor and the magnitude of the observed difference in exchange will surpass any effect of the point mutation. However, in some cases, for example proline mutations, the number of amides will be affected and therefore have a larger effect.

This paper has a description of how peptides with divergent sequences can be handled for comparative HDX studies: doi: 10.1007/s13361-016-1365-5

Reviewer #1 (Remarks to the Author):

In the manuscript titled “Structural dynamics in the evolution of 1 SARS-CoV-2 spike glycoprotein”, Calvaresi et al. utilize HDX-MS to study the structure and dynamics of the SARS-CoV-2 spike protein. They relate their observations to published structures of the spike.

Several of the results presented are consistent with expectations based on existing structural data and/or add to our understanding of the SARS-CoV-2 spike structure and dynamics. These include the increase in HDX in regions of the spike where increased flexibility/mobility is expected because of the D614G mutation, the decreased HDX in the regions spanning the ACE2-RBD interface, ACE2-induced changes in the S2 subunit, and the observation of increased inner flexibility of the Omicron spike RBD that agrees with observations of differing thermostability of the Omicron RBDs ([https://www.cell.com/cell-reports/fulltext/S2211-1247\(22\)00798-7](https://www.cell.com/cell-reports/fulltext/S2211-1247(22)00798-7)).

There are other observations, however, that do not agree with structural data despite the authors’ claims that they do. Notable among these is the observation, which is also a primary result/conclusion of this study that the “substitutions in omicron spike lead to predominantly closed conformations, presumably enabling it to escape antibodies”. The constructs studied in this paper include the 2P mutations. Published cryo-EM structures of Omicron spikes that include the 2P mutations show a preponderance of open states. Two examples of these are:

<https://www.nature.com/articles/s41467-022-28882-9>

<https://www.sciencedirect.com/science/article/pii/S2211124722001528>

The HDX results presented do agree well with structural results obtained without the 2P mutations as described in these papers (which, by the way, are not cited):

[https://www.cell.com/molecular-cell/fulltext/S1097-2765\(22\)00266-0](https://www.cell.com/molecular-cell/fulltext/S1097-2765(22)00266-0)

[https://www.cell.com/cell-reports/fulltext/S2211-1247\(22\)00798-7](https://www.cell.com/cell-reports/fulltext/S2211-1247(22)00798-7)

Indeed, the agreement is quite striking not only with the results related to the higher proportion of the closed conformation, but also the changes in the S2 region that pre-dispose the Omicron spikes to undergo structural changes required for fusion. Bottomline, the differences observed between 2P and non-2P Omicron spikes, especially related to proportions of closed vs open conformations, are now well documented, and the results presented here, agree more with what was observed in non-2P spikes, although the spikes used for the HDX-MS experiments include the 2P mutations. This is an important discrepancy that must be addressed.

We thank the reviewer for pointing out this discrepancy, as indeed, in the interest of saving space, we oversimplified this issue. Following the reviewer’s suggestion, we have now addressed it in the lines 435-441 and 445-446 of the discussion session.

Other issues:

The title of figure 1 “Mechanism of transition from closed to open states” is misleading and over-reaching. This should be rephrased to indicate what the figure is showing, ie., Differences in HDX between Wuhan D614 and G614 spikes.

We thank the reviewer for the suggestion. We have now changed the figure caption in 'Differences in structural dynamics between Wuhan (D614) spike and G614 spike'

Line 150: "Hence, in all analyses we used a stabilized version of spike ectodomains containing '2P' mutations, which make them furin-uncleavable and unable to transition to the postfusion conformation." The 2P mutations do not have the spike furin uncleavable, the RRAR to GSAS substitution in the SD2 subdomain does.

We now changed this sentence to: *'Hence, in all analyses we used a version of spike ectodomains stabilized by R682S, R685S substitutions in the polybasic cleavage sites and K986P, K987P (2P) substitutions, which combined make the spikes furin-uncleavable and unable to transition to the postfusion conformation.'* (lines 143-147)

In Figure 2, panel A, "D614 vs G614 (closed state)" not clear what the "closed state" here indicates.

We have now removed a reference to the 'closed state' in the figure.

Reviewer #2 (Remarks to the Author):

This manuscript describes H/D Exchange of the intact COVID spike protein, comparing all major variants of concern for shifts in conformational dynamics associated with 'open/closed' and 'fusion priming'. The paper is well written, the data are well presented and the results provide some important insights about mutation-dependent dynamic shifts in the NTD, Ace2 binding and fusion-priming. The evidence unambiguously supports the conclusions. I have only a few minor suggestions:

1. The Authors could do a better job of citing previous HDX studies on COVID spike, some of which provide highly relevant foundation and corroborating evidence for the current work. In particular, the work of Ganesh Anand should be cited. Also, the Wilson and Komives groups have measured dynamics of Spike by HDX in various contexts that may be relevant.

We thank the reviewer for the suggestions. We had cited two of Ganesh Anand's papers (ref. 35 and 39), and we have now included his recent preprint (ref. 40). We have also now included the work for Wilson group on the RBD (ref. 41) and a very recent article based on mass photometry data (ref. 42). However, we could not find any HDX-MS study on spike conducted by Komives group.

2. The 'bubble plots' are an excellent way of representing EX1 kinetics. It would be interesting to see these for more than one peptide in this region, perhaps in supplemental. To have full confidence in these data, similar kinetics for overlapping or nearby peptides should be included.

We thank the reviewer for this suggestion. We have now analysed with HX-Express2 another peptide spanning the RBM (peptide 495-510) and included this extra analysis in the supplementary information (a peptide spanning the same residues is unfortunately absent in the omicron spike) (Fig. S44-45). It is worth noting that the HX-express fitting returned very similar relative size of populations and their time-dependent evolution, for every spike variant, as the peptide previously included. We have also extracted

from DynamX the stacked spectral plots of many other peptides encompassing this region, and included them in the supplementary figures (Fig. S41). We included peptides of both high and moderate data quality, to demonstrate that bimodal spectra, or peak broadening, were visually distinguishable across all of them, regardless the peptide signal-to-noise ratio. Here below the additional figures.

Supplementary Fig. 41 (a,b). EX1 kinetics in selected peptides spanning residues 495-503 of spike receptor binding motif (RBM) of spike trimers and the isolated RBD in complex with ACE2. Stacked spectral plots elucidate the evolution of the bimodal isotopic envelopes across the time points studied and for different overlapping peptides in spike trimers. The bimodal isotopic envelopes of peptides 495-510 and 495-513 have been also analysed with HX-Express² (see Supplementary Figs. 42-45). The isotopic envelopes of the isolated ancestral RBD does not display bimodal features nor evident peak broadening.

Supplementary Fig. 44. EX1 kinetics in the receptor binding motif (RBM) of spikes in complex with ACE2. From top to bottom: peptide YGFQPTNGVGYQPYRV (495-510) of Wuhan spike; peptide YGFQPTNGVGYQPYRV (495-510) of G614 spike; peptide YGFQPTNGVGYQPYRV (493-509) of delta spike; peptide YGFQPTYGVGYQPYRV (492-507) of alpha spike; peptide YGFQPTYGVGYQPYRV (492-507) of beta spike. From left to right: bubble plots representing the level of deuteration and the relative intensity of the low- and high-mass envelope (binomial fit 1 and binomial fit 2, respectively) for bimodal isotopic distributions and the level of deuteration of the unimodal envelope (unimodal fit), set at 100% intensity, in the ACE2 bound state; peak width of the isotopic distribution (calculated at 20% of Baseline Peak Intensity -BPI) and number of exchangeable amides (#NHs), in the bound state; deuterium level of the low-mass envelopes (binomial 1), high-mass envelopes (binomial 2) and unimodal distributions (centroid) in ACE2-bound spike; deuterium level of spikes in apo state (unimodal distributions – centroid). Spectral analysis was conducted with HX-Express2² and graphs were automatically generated upon fitting the isotopic envelopes shown in Supplementary Fig. 45. Bimodal fit was applied to peptide spectra at time points showing clear features of bimodality and/or enhanced peak width. Note that the high-mass population does not manifest at early time points, thus does not follow the HDX of the unbound state.

a

b

Supplementary Fig. 45. EX1 kinetics in the receptor binding motif (RBM) of spikes in complex with ACE2. Mass spectra of the representative peptides of the RBM (listed in Supplementary Fig. 44) deconvoluted with bimodal or unimodal fittings with HX-Express 2. A) Mass spectra of peptide of Wuhan, G614 and delta spikes. B) Mass spectra of peptide of alpha, beta and omicron spikes. The horizontal orange lines indicate the distribution width at 20% BPI; the red dots mark the envelope peaks; the vertical green lines indicate the centroid mass; the dark blue lines fit the unimodal envelope distributions deconvoluted with binomial fitting; the green lines fit the low-mass envelope distributions deconvoluted with bimodal fitting; the light blue lines fit the high-mass envelope distributions deconvoluted with bimodal fitting.

3. The idea of increased dynamics as a way of evading antibody binding is an interesting one. The authors may wish to reflect on how this mechanism would impact a continuous vs. discontinuous epitope in the discussion. (At first pass, my thought would be that this mode of evasion would be much more impactful on discontinuous epitopes than on continuous ones, and that it would never offer 'complete' protection, since the 'vulnerable' configuration would still occasionally be populated... Essentially, manipulation of conformer selection for antibody binding).

This is a very interesting input and we completely align with the reviewer's thoughts. We have added the following paragraph in the discussion (lines 410-415):

"The conformational plasticity of this NTD site presumably allows antibody-vulnerable configurations of conformational epitopes to remain occasionally populated, reducing, but not completely abrogating, antibody-mediated protection. At the same time, linear NTD epitopes are likely more impacted by the residue changes in spike variants rather than by a different conformational aspect of the NTD of VOCs."

Reviewer #3 (Remarks to the Author):

The manuscript by Calvaresi reports an extensive HDX-MS comparative analysis of various CoV2 spike proteins in the unbound or ACE2-complexed states. The data are well presented, and the study can reveal some useful insight into the solution behavior of the spike protein. I applaud the authors for using freshly purified proteins to minimize confounding effects of freeze thaw that have previously been observed. The methods and the majority of the analysis is solid, including the direct comparison of the solution behavior of all the variants. There are a few major concerns going back to the experimental design for the ACE2 bound complexes as outlined below, along with a other minor issues listed below that need to be addressed.

Major concern:

Why did the authors use a ratio of 1:2 spike:ACE2? Won't this mean that 1 of the 3 RBDs of the spike is unbound? For the most conclusive HDX studies it is really important to favor full binding of your protein of interest so that you measure the full extent of changes associated with binding. With the 1:2 ratio used here, only a maximum of 66% of the spike RBD population can be in the bound form. This may manifest as two apparent populations for regions that are most perturbed by ACE2 binding.

This an important point and we thank the reviewer for the opportunity to clarify it. For a Spike trimer to be ACE2 binding-competent at least one of its RBDs must be erect and a mixture of its open and closed states within the trimer is well known in the literature (see e.g. <https://doi.org/10.1016/j.cell.2020.02.058>, <https://www.nature.com/articles/s41594-020-0468-7.pdf?origin=ppub>, <https://www.pnas.org/doi/10.1073/pnas.2022586118>, and <https://www.nature.com/articles/s41467-022-28882-9>). Spikes with all three RBDs accessible are rarely observed, at least in cryoEM. Although we cannot be sure of the precise ratios of up and down states in spikes in solution, in non-cryogenic state, the conformational dynamics between these adopted structures governs this interaction and is what our solution HDX-MS experiments can report on. In our experiments we, therefore, try to capture a situation where we can monitor this behaviour within the experimental limitations of required sample amounts and concentration – with large concentrations of the material difficult to produce.

The 1:2 ratio was chosen after initial optimization HDX-MS experiments. We incubated Wuhan and G614 spikes at 1:2 and 1:3 ratio spike:ACE2 and found the same difference in HDX for spike peptides within the RBM at both ratios. This suggests that increased ACE2 above 1:2 was not increasing spike binding occupancy, but likely leads to an increased population of unbound ACE2 in our labelled sample. Accordingly, the HDX differences observed within ACE2 incubated at 1:2 and 1:3 ratios spike:ACE2 appeared significantly higher at ratio 1:2 than at 1:3, which supports that a higher fraction of unbound ACE2 is present in the latter case. In support of the ratio chosen, a very recent mass photometry paper (<https://pubs.rsc.org/en/content/articlelanding/2022/cc/d2cc04711j>) on Wuhan and omicron spikes (in the same '2P' pre-fusion stabilised version we use) reported that 1:4 spike:ACE2 shows predominantly spike bound to one ACE2, with some spike bound to 2 ACE2 but negligible amounts bound to 3 ACE2.

Considering these observations, 1:2 ratio likely provides information on saturated spike ACE2-accessible 'up' states. As the reviewer points out, this would mean that one of the RBDs will be unbound, but it is likely to be in an ACE2-inaccessible "down" state. The experiment performed to address major concern 2 provides further evidences in support of the ratio chosen, for every spike analysed (please see below).

In this revised version, we inserted this optimization procedure in a dedicated paragraph of the method session (*Optimization of the HDX conditions – lines 546-554*) and added two figures showing these results in the supplementary information and present them below.

Fig. S47. Histogram illustrating the magnitude of decreased HDX (Δ HDX) manifesting in selected peptides of the RBM when Wuhan and G614 spikes are incubated at 1:2 and 1:3 ratios with ACE2. No significant differences in Δ HDX are observed between the two incubation ratios and for both spike trimers.

Supplementary Fig. 48. Histogram illustrating the magnitude of decreased HDX (Δ HDX) manifesting in selected peptides of the ACE2 binding site when ACE2 is incubated at 2:1 and 3:1 ratios with Wuhan spike. A minor decrease in HDX is observed at ratio 3:1.

Were bimodal spectra observed across different regions of the spike beyond the 495-513 region that was presented?

Bimodal spectra were not observed in apo spikes by working with freshly prepared spike samples. In the ACE2-bound state, beyond the 495-513 region, we observed - for every spike trimer analyzed - the appearance of bimodal isotopic envelopes in the HR1 region spanning residues 962-982, which is one of the segments becoming more dynamic during the priming for fusion of spike. In the interest of writing the manuscript for a broad readership, we did not describe the specific HDX regime of this particular segment in detail, rather summarizing the increased HDX observed in various S2 stretches as 'increased dynamics'. However, following the reviewer's inquiry, we have reconsidered our initial thoughts and have now described the HDX bimodality seen in this region of bound-spikes in the manuscript text, as presumably it also contributes to the priming mechanism (lines 355-363).

"Notably, peptides spanning residues 962-982 manifested increased HDX in the form of bimodal isotopic envelopes when spikes are engaged to ACE2 (Fig. 6c and Supplementary Fig. 46). This segment encompasses the HR1 helix, which undergoes a large conformational rearrangement upon the transition to post-fusion state¹⁵, suggesting that our analysis captured the specific dynamic events leading to the HR1 reorientation, which primes spikes for fusion. The breadth of this HDX bimodality varies with omicron > alpha > beta > G614 ≈ Wuhan spike, with omicron spike pre-manifesting it also in the absence of ACE2 (Fig. 6c and Supplementary Fig. 46)."

Additionally, we have modified Fig. 6c in the main text to include the isotopic envelopes of peptides 963-977, in place of its uptake plots. We believe that the envelopes deliver a much clearer message. We also added a supplementary figure (S46) including the stacked spectral plots of peptide 963-977 and an overlapping peptide (962-977).

Fig. 6 Spike priming for fusion. A) Regions manifesting increased dynamics in spikes bound to ACE2 are superimposed and colored in red on a single protomer of the structure of D614 spike with one RBD bound (PDB: 7a95¹⁴); regions with increased dynamics only in spike of alpha and omicron variants bound to ACE2 are colored in magenta. B) The magnitude of the destabilization of the core helices is represented by differential colouring (red scale) for the various spike trimers. The HR1 region (962-982) manifesting HDX bimodality is framed in grey. C) The bimodal isotopic envelopes of a model peptide spanning the HR1 region 962-982 are shown at 15 s and 1 min time points for spike (S) variants in the ACE2-bound form and for omicron spike alone, to exemplify their priming for fusion upon receptor engagement. Bimodal envelopes manifest in omicron spike also in the absence of ACE2, indicating it as pre-primed for fusion.

Supplementary Fig. 46. HDX bimodality in the HR1 of spikes in complex with ACE2 and omicron spike alone. Stacked spectral plots of peptides 962-977 and 963-977 illustrate the evolution of the bimodal isotopic envelopes over the time points studied. Particularly, time points 15 s and 1 min (23 °C) show clear separation between the low- and high-mass envelopes in every spike trimer, with the relative intensity and centroids of the two envelopes reporting on the degree of destabilization exerted by the ACE2 binding on the HR1. The isotopic envelopes in omicron spike alone manifest evident peak broadening, suggesting that the its HR1 follows a bimodal HDX behaviour even in the absence of ACE2.

Additionally, how long was the spike:ACE2 complex incubated prior to exchanges? Do the authors know if this time was sufficient for equilibration to the 66% bound form?

Spikes-ACE2 complexes were incubated for one hour before starting the deuterium labelling (we now specified this in the method session). Given the Kds are, depending on the variant, in the order of 10 to 150 nM with k_{ons} between 0.07 and 0.2 $\mu\text{M}^{-1}\text{s}^{-1}$ (<https://doi.org/10.1038/s41467-022-28768-w> - see supplementary info), we believe that this incubation period largely suffices for system equilibration.

Bimodal spectra at the 495-513 region presented look convincing. Based on the authors' coverage maps each of the mutants have several peptides that span this same region in the spike proteins. The authors need to check and make sure that the bimodal (or peak broadening) is evident in all of those overlapping peptides too. I do not think it is critical for the authors to fit and thoroughly analyze all of the overlapping peptides, as it is not a central conclusion of the paper, but at the least the authors need to validate that it is observed consistently among the overlapping peptides and make a note of this in the results.

We thank the reviewer for highlighting this important point. We have now extracted from DynamX the stacked spectral plots of many other peptides encompassing this region and included them in the supplementary figures. We included peptides of both high and moderate data quality, to demonstrate that bimodal spectra, or peak broadening, were visually distinguishable across all of them, regardless of the peptide signal-to-noise ratio (Fig. S41). We also analysed the bimodal isotopic distributions of another peptide (495-510) with HX-express2 and included this extra analysis in the supplementary information (a matching peptide is unfortunately absent in the omicron variant) (Fig. S44-45). It is worth noting that the HX-express2 fitting returned very similar relative size of populations and their time-dependent evolution, for every spike variant, compared to the peptide previously included. The result of this extra analysis could also address the reviewer's following concern about the calculated size of the populations.

Supplementary Fig. 41 (a,b). EX1 kinetics in selected peptides spanning residues 495-503 of spike receptor binding motif (RBM) of spike trimers and the isolated RBD in complex with ACE2. Stacked spectral plots elucidate the evolution of the bimodal isotopic envelopes across the time points studied and for different overlapping peptides in spike trimers. The bimodal isotopic envelopes of peptides 495-510 and 495-513 have been also analysed with HX-Express² (see Supplementary Figs. 42-S45). The isotopic envelopes of the isolated ancestral RBD does not display bimodal features nor evident peak broadening.

Supplementary Fig. 44. EX1 kinetics in the receptor binding motif (RBM) of spikes in complex with ACE2. From top to bottom: peptide YGFQPTNGVGYQP_YRV (495-510) of Wuhan spike; peptide YGFQPTNGVGYQP_YRV (495-510) of G614 spike; peptide YGFQPTNGVGYQP_YRV (493-509) of delta spike; peptide YGFQPTYGVGYQP_YRV (492-507) of alpha spike; peptide YGFQPTYGVGYQP_YRV (492-507) of beta spike. From left to right: bubble plots representing the level of deuteration and the relative intensity of the low- and high-mass envelope (binomial fit 1 and binomial fit 2, respectively) for bimodal isotopic distributions and the level of deuteration of the unimodal envelope (unimodal fit), set at 100% intensity, in the ACE2 bound state; peak width of the isotopic distribution (calculated at 20% of Baseline Peak Intensity -BPI) and number of exchangeable amides (#NHs), in the bound state; deuterium level of the low-mass envelopes (binomial 1), high-mass envelopes (binomial 2) and unimodal distributions (centroid) in ACE2-bound spike; deuterium level of spikes in apo state (unimodal distributions – centroid). Spectral analysis was conducted with HX-Express2² and graphs were automatically generated upon fitting the isotopic envelopes shown in Supplementary Fig. S45. Bimodal fit was applied to peptide spectra at time points showing clear features of bimodality and/or enhanced peak width. Note that the high-mass population does not manifest at early time points, thus does not follow the HDX of the unbound state.

a

b

Supplementary Fig. 45. EX1 kinetics in the receptor binding motif (RBM) of spikes in complex with ACE2. Mass spectra of the representative peptides of the RBM (listed in Supplementary Fig. 44) deconvoluted with bimodal or unimodal fittings with HX-Express 2. a) Mass spectra of peptide of Wuhan, G614 and delta spikes. b) Mass spectra of peptide of alpha, beta and omicron spikes. The horizontal orange lines indicate the distribution width at 20% BPI; the red dots mark the envelope peaks; the vertical green lines indicate the centroid mass; the dark blue lines fit the unimodal envelope distributions deconvoluted with binomial fitting; the green lines fit the low-mass envelope distributions deconvoluted with bimodal fitting; the light blue lines fit the high-mass envelope distributions deconvoluted with bimodal fitting.

While the peak broadening is evident by eye, it also looks like the separation of the two populations within the bimodal is poor. Because of this, I do not think that there is high confidence associated with the fitting to extract the exact deuterium incorporation and exact sizes for the two populations. The authors interpret the bimodal as EX1 kinetics, but the two populations could easily result from the 66% that is ACE2 bound and the remaining 33% that is unbound. There will likely also be a distribution of different stoichiometries of ACE2-spike among the different spike molecules. For example there will be some finite number of spikes with all three lobes bound to ACE2, the majority with 2 lobes bound, a small fraction of only a single lobe bound, and a really small population of completely unbound spike. These different populations can easily result in complicated bimodal mass envelopes that have nothing to do with true EX1 kinetics. One thing the authors can do is to see if the higher deuterated population appears consistent with what was observed in the unbound spike. If bimodal profiles are observed in both the spike and the ACE2 then the authors might be able to interpret what fraction of the protein was actually bound.

These issues could have been alleviated if the experiment was designed with a sufficient excess of ACE2 to ensure fully bound complexes. I appreciate that these reagents are challenging to make and the experiments are difficult to carry out, and do not expect the authors to go back and redo the experiment. Furthermore, I think the data in this section can still be insightful, but the authors need to reexamine and rewrite this portion with full acknowledgment of all the confounding factors that limit how much can be concluded. The problem of incomplete ACE2 binding may also be a confounding factor for the following section on the S2 subunit priming.

We thank the reviewer for considering this important aspect. We based our interpretation on the following observations and experiments.

As the reviewer suggests, we had carefully compared the HDX of the high-mass population of the bound states and the HDX of the population (unimodal spectra) of the apo state (also shown in fig. S42 and S44). The two HDX profiles do not appear consistent. We believe that, if the ACE2 was not sufficient to saturate the binding of the monomers with accessible (erect) RBD, the high-mass population in the bound states should align on the m/z scale to the apo spike states, manifesting already, and with clear separation from the low-mass population, at 4 s or 20 s on ice, but this does not appear to be the case. We do not observe it at 4 s on ice for Wuhan and G614 bound-spikes, or at 20 s on ice for alpha, beta and omicron bound-spikes. We have now added a note on this in the figure caption of fig. S42 and S44.

To better investigate this aspect, for this revised version, we performed an extra experiment. We performed HDX-MS on the isolated RBD of the ancestral Wuhan spike in the presence and absence of ACE2, at ratio 3:2 RBD:ACE2, which simulates our 1:2 spike trimer:ACE2 ratio used in our experiments. Given that the whole population of isolated RBD molecules are binding-competent, with such a binding stoichiometry, 33% of the RBD population remain effectively unbound in the presence of ACE2. We did not observe bimodal distributions in the RBM of the bound-RBD state, indicating that a mixture of bound and unbound monomers, with same conformational characteristics in the apo state, does not manifest with a clear envelope bimodality under the conditions studied. We now included the extracted stacked spectral plot of the bound-RBD in the supplementary figures (please, see previous figure S41). Furthermore, the preliminary data shown above and the experiment performed to address major point 2 support a scenario where spike erect RBDs are saturated in binding occupancy.

These observations prompt us to associate the RBM bimodal HDX profiles to the spike cooperative binding mode reported in previous studies; we thus interpreted the high-mass population as an extra monomer that erects upon ACE2 binding (in a cooperative manner) and is likely able to engage with an extra ACE2 molecule, thus displaying an HDX profile that differs from the monomers in the unbound state. This gives rise to a fine-tuned equilibrium of states, characteristic for each spike variant, that we aimed at deciphering with our analysis.

We agree with the reviewer that we should not strictly refer to this HDX behaviour as EX1 kinetics, as we associated it to the behaviour of different monomers, and not to an individual protein stretch displaying correlated exchange. We thus now refer to it as “HDX bimodality”. We also modified the text (lines 306-324) to better explain these observations and include the extra experiment performed, and we present it here below.

“Furthermore, we observed that the HDX profiles of all peptides spanning the RBM of spike trimers (residues 495-503) in the ACE2-bound states showed bimodal isotopic distributions, hence a high- and a low-mass population, whereas a single unimodal distribution characterized the apo states (Fig. 5 and Supplementary Figs. 41-45). The HDX of the high-mass populations in the bound states appeared inconsistent with the HDX of the respective apo states. In contrast, the ACE2-bound state of the isolated ancestral RBD (3:2 RBD: ACE2), containing a significant fraction (33%) of unbound population, did not display bimodal isotopic distributions in the RBM (Supplementary Fig. 41), ruling out that a mixture of bound and unbound populations, with same conformational characteristics in the apo state, manifests with an HDX bimodality under the conditions studied. These data thus suggest that the RBD of bound spikes can explore two distinct and slowly interconverting populations, which exchange giving rise to two resolved isotopic distributions. We rationalize, based on the receptor binding mode reported in previous studies^{14,55}, that the bimodal HDX profile of the bound spike states reports on cooperative opening within the spike trimer, with the less exchanged (low-mass) population accounting for open protomers with a bound RBD, whilst the more exchanged (high-mass) population likely corresponding to closed protomers transitioning to the open state and readying to engage another receptor molecule.”

We also would like to highlight that bimodal spectra were not observed in the bound states of ACE2, even though we observed these states at uncomplete binding occupancy (please, see answer to Major point 2).

Major point 2:

The authors examine HDX changes within ACE2 in complex with the various spike constructs to assess the binding affinity and avidity. This section is highly problematic. I am having trouble trying to understand the logic of why this experiment would reveal what the authors suggest. I agree that tighter binding will lead to more protection in the ACE2. Based on the methods the authors preincubated at a ratio of 1:2 spike trimer:ACE2. This ratio should correspond to a 3:2 ratio of spike monomer:ACE2, and if the interaction is of sufficient affinity then the vast majority of the ACE2 should be bound. For HDX studies you typically want an excess of the ligand with sufficient preincubation so you can be sure that you have near-full occupancy of the binding sites in the protein of interest. If not then you risk looking at a mixture of bound and unbound species that will depend on affinity and possibly incubation time. Depending on the association/dissociation kinetics this can also result in bimodal isotopic envelopes as the data might reflect a combination of free and bound populations, further complicating the analysis.

The authors suggest that since previous binding studies have indicated similar affinities between WT and G614, that a higher portion of the ACE2 is bound in the presence of G614 spikes. While this may be true, this would indicate that the complexes were not incubated long enough to reach equilibrium. At equilibrium with equal affinities there should be an identical amount of complex formed for both WT and G614 spikes. If there isn't then it might relate to the formation of the ACE2-WT complex being much slower, perhaps because more time is needed for the WT to sample open conformations capable of binding.

The reviewer is right; however, the experiments we referred to were all based on methods (biolayer interferometry) that measure only the affinity of a single monomer binding to one ACE2, as spikes are immobilised and thus their concentrations (as well as the effective concentrations of individual accessible RBDs) irrelevant for the measurement, while ACE2 monomers form a mobile phase. We have now specified this in the text (lines 269-270). In our HDX-MS experiments, we equilibrated all complexes for one hour before labelling, which, considering the aforementioned favourable k_{on} and K_d , should allow for equilibration to be reached.

Additionally, if the binding is incomplete then I would expect to see a bimodal isotopic profile near the binding site reflecting the population of bound and unbound ACE2. This bimodal may actually be a much more direct way to assess what portion of the ACE2 is able to bind the spike. However, in many cases the two populations in bimodal spectra are not well-resolved and quantifying the populations is difficult to determine with any confidence. Overall, this assay has a lot more variables and caveats than the authors account for, and I would be very hesitant to conclude anything from this data. I'm guessing the authors thought they could extract some additional information for the ACE2 since it's already in the dataset, but this is not a well-designed experiment for any solid conclusions. It might be a useful assay for future studies to examine kinetics of binding among the variants, but since this is not an integral part of the current paper, I advise the authors simply omit

We thank the reviewer for highlighting this important point and below we provide further details to support our experimental design and their conclusions.

Despite the molar excess of spike monomers, ACE2 is in excess compared to the binding-competent open monomers with available erect RBDs within spike trimers. To confirm this also in HDX-MS, we performed an additional experiment for this revised version of the manuscript. We labelled ACE2 in the presence and absence of the isolated ancestral RBD, at a 3:2 ratio RBD:ACE2, which simulates a 1:2 spike trimer:ACE2 ratio. Differently from the RBDs embedded in a spike trimer, the whole population of isolated RBD is binding-competent. In this scenario, the RBD is effectively in excess compared to ACE2, granting full ACE2 binding occupancy, and indeed, we observed that the binding effect on ACE2 manifested with much greater magnitude (cumulative $\Delta\text{HDX} = 19.15$ Da) compared to that induced by spike trimers, which show generally less cumulative ΔHDX due to the presence of a (varied) fraction of unbound ACE2 in the bound state. This experiment indicates that the ACE2 is not saturated in binding occupancy in the HDX-MS experiments performed on spike trimers (hence spike trimers are saturated or close to saturated), therefore we argue that the magnitude of HDX effects (ΔHDX) on the ACE2 holds useful information on the spike-ACE2 binding stoichiometry.

It has to be noted that the ΔHDX induced by alpha, beta and delta spikes also have a contribution from the stability of the hydrogen bonding network engaged with ACE2 (related to the affinity of their individual binding-competent monomers to the receptor), which will result in a cumulative effect given by their binding stoichiometry and affinity (reported as higher than for G614 in several studies). This is presumably the reason why for alpha spike a cumulative ΔHDX slightly higher than that of the isolated RBD was observed.

In our hands, the incomplete binding occupancy of ACE2 did not manifest with isotopic bimodal distributions in its spike binding sites, making it difficult to estimate the fraction of unbound ACE2 for the different trimers. This is one of the reasons why we refrain from proposing any stoichiometry model.

We have now included the results of this extra experiment (see figures below) in fig. 3 of the main text and Fig S27 and S40 of supplementary information and have also rewritten this section in the main text to provide a clearer explanation of our rationale (lines 243-255).

“Next, by studying the HDX of the ACE2 ectodomain alone and in complex with spike trimers (1:2 spike trimer:ACE2) and the isolated ancestral RBD (3:2 RBD:ACE2), we measured the magnitude of the HDX effects (ΔHDX) induced by spike binding to ACE2. The whole population of the isolated RBD is binding competent, granting complete occupancy of the ACE2 binding sites, whereas only a fraction of the RBDs embedded within spikes are erect and thus able to engage the receptor. The observed ΔHDX results from a cumulative effect of binding stoichiometry (how many ACE2 molecules are bound) and the stability of the hydrogen-bonding network between spikes and ACE2 (which can be related to the spike-receptor binding affinity), enabling us to rank the spike-receptor binding avidity (i.e. the overall strength of binding arising from the affinity of an individual RBD-ACE2 interaction and the stoichiometry of each spike trimer engaging between zero and three ACE2 molecules at once).”

For clarity, we also modified the following sentence (lines 261-268):

“The cumulative ΔHDX induced by different spikes varied with $\alpha > \beta > \delta > \text{G614} > \text{Wuhan} \approx \text{omicron}$ (Fig. 3b and Supplementary Fig. 28). These ΔHDX values were generally lower than that induced by the isolated RBD, indicating that a fraction of ACE2 molecules remained unbound in the spike:ACE2 states, thus suggesting that all binding-competent RBDs within the trimers were fully occupied.”

Fig. 3 Effect of spike binding on ACE2 dynamics. A) Regions of ACE2 manifesting a significant decrease in HDX upon spike binding are superimposed on the structure of ACE2 ectodomain bound to RBD (PDB: 2ajf), colored in blue scale according to the magnitude of the HDX effect. The region colored in red indicates increased HDX upon binding, in dark gray regions with no coverage. B) ACE2 binding avidity. The cumulative difference in HDX (Δ HDX) between ACE2 alone and ACE2 bound to spikes and the isolated ancestral RBD for selected peptides spanning binding sites and across time points 20 s on ice, 10 min at 23 °C and 360 min at 28 °C is plotted. A plot for all time points in Supplementary Fig. 28.

Supplementary Fig 27. Difference plot illustrating the difference in HDX between ACE2 in complex with the isolated ancestral RBD and ACE2 alone (orange line indicates: 20 s on ice, green line: 10 min at 23 °C, dark blue line: 360 min at 28 °C). Peptide segments of interest are highlighted. The peptides are arranged according to their peptide centre residue. A dotted grey line indicates the 98% CI as a threshold for significance and a dotted black line the 99% CI as a threshold for significance.

Supplementary Fig 40. Difference plot illustrating the difference in HDX between the isolated ancestral RBD in complex with ACE2 and the isolated ancestral RBD alone (orange line: 20 s on ice, green line: 10 min at 23 °C, dark blue line: 360 min at 28 °C). Residues comprising a region with significant differences in HDX are indicated. The peptides are arranged according to their peptide centre residue. A dotted grey line indicates the 98% CI as a threshold for significance.

Minor point:

The authors should make a note that there is a caveat when comparing peptides among spike variants where the peptide sequences are not perfectly matched. The intrinsic rates of exchange will be offset by some of the amino acid substitutions, so it is possible to observe different exchange kinetics even if there is no actual difference in the structural dynamics of that region of the protein. The authors should make a note how they handled the comparison at regions where peptide sequences were not matched. It should help that the authors had a fully deuterated control so they can compare % deuteration instead of deuterium uptake, but this won't alleviate the caveat entirely. For example, one of the largest changes seen in with beta spikes was attributed K417N, and from the coverage map it looks like all of the peptides reporting around this region span residue 417. The authors should check how much of an effect the mutation will have on the intrinsic exchange rate. In many cases it will be relatively minor and the magnitude of the observed difference in exchange will surpass any effect of the point mutation. However, in some cases, for example proline mutations, the number of amides will be affected and therefore have a larger effect.

This paper has a description of how peptides with divergent sequences can be handled for comparative HDX studies: doi: 10.1007/s13361-016-1365-5

We gratefully thank the reviewer for highlighting the caveat arising from the comparison of peptides harboring mutations and suggesting the paper (which we now cited). We briefly explained in the method section how they were compared but did not consider this to a sufficient degree. We explain our comparative workflow below and the additional k_{ch} considerations done:

Workflow:

We selected for comparison only peptides with identical cleavage, i.e. same N- and C-termini. Thanks to the numerous peptides available and the high redundancy, we could, in most instances, find matching peptides. However, this became impossible in case of deletions and insertions. We normalized the uptake values of mutant peptides by the uptake of their MaxD (= fully deuterated control) and obtained absolute uptake values (in Da) referencing to G614 spike, with the following equation (now included in the method session):

$$\Delta HDX = \left(\frac{DU \text{ mutant peptide}}{DU \text{ MaxD mutant peptide}} \times DU \text{ MaxD G614 spike peptide} \right) - DU \text{ G614 spike peptide}$$

For instance, at a given time point, peptide X of G614 spike has DU of 3 Da with a MaxD of 10 Da; its matching peptide Y in alpha spike has DU of 4 Da with MaxD of 10.3 Da; the normalized DU for alpha spike is 3.88 Da. The ΔHDX reported in the butterfly plot is +0.88 Da. This was done with the aim to insert mutant peptides in the butterfly plots, making the HDX comparison more visually intuitive and easily readable.

Considering k_{ch} differences:

We acknowledge the fact that the MaxD does not entirely alleviate the caveat arising from the difference in k_{ch} between peptides harboring mutations. Before performing the experiments, we had checked the values of k_{ch} (according to the 2018 updated values: doi: 10.1007/s13361-018-2021-z) and

noticed that the differences are minor and most likely give rise to ΔHDX below the threshold of significance when comparing peptides without a significant difference in k_{op} .

However, we now performed a more thorough analysis on the impact that the k_{ch} offset has on the differences measured. We included this analysis in a supplementary table (Supplementary Data 1). We tested a null hypothesis in which every difference in HDX observed arises from a difference in k_{ch} .

In more detail: we calculated the average k_{ch} for a peptide and its mutant variant - excluding the N-terminal residue (one peptide per mutation was analyzed). The average k_{ch} was selected as the individual amide HDX rates are averaged when measured by MS at peptide level. We then calculated the % of observed ΔHDX for that peptide normalized by MaxD, selecting the time point showing the highest ΔHDX , as considered the most sensitive to differences.

In most instances, as the reviewer foresaw, the % ΔHDX significantly surpasses the % Δk_{ch} , including for peptides spanning K417N (the table includes a 'note' column describing the outcome of the analysis). Therefore, while acknowledging the fact that the ΔHDX values have an offset at quantitative level, the observed ΔHDX can be considered qualitatively reliable. Only for one peptide (946-961 of delta spike), we cannot unambiguously demonstrate that the observed ΔHDX arises from a real difference in dynamics in respective to G614 spike. Therefore, we have not based any discussion on that peptide.

We now mentioned this in lines 184-187 of the results and included the description of this approach in the method session (lines 612-630), acknowledged the presence of this caveat. We also included new figures in the supplementary information (Fig. S4) where residue-level k_{ch} ratios are plotted.

“To compare peptides containing residue substitutions in spike variants (mutant peptides) with peptides of G614 spike, segments with identical N- and C-termini were selected. Their difference in deuterium incorporation (ΔHDX) was calculated according the equation 1 and plotted in Supplementary Figs. 9-12:

$$\Delta\text{HDX} = \left(\frac{\text{DU mutant peptide}}{\text{DU MaxD mutant peptide}} \times \text{DU MaxD G614 spike peptide} \right) - \text{DU G614 spike peptide} \quad (1)$$

To estimate the impact of the difference in chemical exchange rate constants (k_{ch}) on the observed ΔHDX between mutant peptides and peptides of G614 spike⁵⁰, firstly the k_{ch} of individual residues within the spike protein sequences were calculate⁴⁹. Successively, at peptide level, the percentage of difference in k_{ch} (% Δk_{ch}) were compared to the percentage of ΔHDX normalized by the MaxD (% ΔHDX) in the time point showing maximal effect, as reported in Supplementary Data 1. The identified differences in HDX between spike variants and G614 spike in segments spanning amino acid changes resulted of high-confidence, with the impact of k_{ch} negligible, except for peptide 946-961 of delta spike.”

Supplementary Fig. 4. Influence of amino acid changes on the k_{ch} of residues of spike variants. The ratios between the k_{ch} of spike variants and Wuhan spike and the k_{ch} of G614 spike residues is plotted from residue 1 to 628 (a) and from residue 629 to 1256 (b). Values are extracted from Supplementary Data 1. Amino acid changes are illustrated on the left of the graphs.

REVIEWERS' COMMENTS

Reviewer #3 (Remarks to the Author):

The authors have addressed many of the questions I had, and I commend them on including much of the data that is used to inform how they interpret their bimodal spectra. Based on this data and their response, there are a few last minor points that should be addressed prior to publication.

I agree with the authors that the isolated RBD experiment sheds a lot of light on the source of the bimodal spectra, so it likely does not stem from a lobe of the trimer simply being unbound. In line 322, the authors state “whilst the more exchanged (high-mass) population likely corresponding to a closed protomer transitioning to the open state and readying to engage another receptor molecule”. This statement makes it sound like the high-mass population should be able to bind Ace2, but then why is it unable to bind another molecule of Ace2? Maybe rephrasing this line to indicate that the high-mass population is somehow perturbed, but somehow still does not engage Ace2 like the other lobes of the trimer would help minimize confusion. If prior literature has made speculations about what this third lobe could be doing, then I recommend referencing those here.

- The attached additional spectra help confirm the reproducibility of the observed bimodals, but for several cases the two populations are so poorly resolved that deconvolution to extract data specific to each subpopulation can be ambiguous and potentially misleading. For example in figure S45 the spectra for G614 at 15 s and 1 min can likely be fit just as well with many other combinations of deuterium levels and intensities. The specific phrase: “The HDX of the high-mass population in the bound states appeared inconsistent with the HDX of the respective apo states.” should be clarified so that it specifically refers to the earliest time point where no evidence was seen for a population consistent with unbound RBD.

- In light of the attached spectra data there is one other possibility the authors should consider as a source of the observed bimodal spectra. Dissociation of ACE2 during deuterium exchange might also explain observed bimodals presented in Fig S42, S44, as unbinding of ACE2 during D2O incubation will likely start to occur in a matter of minutes. The general trend from Wuhan/G14 to Beta to Alpha showing later transitions in the EX1, appear to match the same trend in koff kinetics reported by Wrobel et al: <https://www.nature.com/articles/s41467-022-28768-w#MOESM1>. This source of bimodal would also be consistent with the earliest time point in the Ace2 bound form not yet showing a second (highly deuterated population) as the dissociation has yet to occur to any appreciable degree. I don't think the authors need to elaborate on this, but I think this is something the authors should at least mention as another possible confounding factor that is influencing the observed bimodal spectra in either the results or discussion.

- The authors should also include what temperature the pre-binding with Ace2 for 1 hour was conducted at. I know this seems nitpicky but there several labs working on similar systems and temperature may drastically affect binding kinetics for anyone attempting to reproduce these studies.

Reviewer #1 (Remarks to the Author):

In the manuscript titled “Structural dynamics in the evolution of 1 SARS-CoV-2 spike glycoprotein”, Calvaresi et al. utilize HDX-MS to study the structure and dynamics of the SARS-CoV-2 spike protein. They relate their observations to published structures of the spike.

Several of the results presented are consistent with expectations based on existing structural data and/or add to our understanding of the SARS-CoV-2 spike structure and dynamics. These include the increase in HDX in regions of the spike where increased flexibility/mobility is expected because of the D614G mutation, the decreased HDX in the regions spanning the ACE2-RBD interface, ACE2-induced changes in the S2 subunit, and the observation of increased inner flexibility of the Omicron spike RBD that agrees with observations of differing thermostability of the Omicron RBDs ([https://www.cell.com/cell-reports/fulltext/S2211-1247\(22\)00798-7](https://www.cell.com/cell-reports/fulltext/S2211-1247(22)00798-7)).

There are other observations, however, that do not agree with structural data despite the authors' claims that they do. Notable among these is the observation, which is also a primary result/conclusion of this study that the “substitutions in omicron spike lead to predominantly closed conformations, presumably enabling it to escape antibodies”. The constructs studied in this paper include the 2P mutations. Published cryo-EM structures of Omicron spikes that include the 2P mutations show a preponderance of open states. Two examples of these are:
<https://www.nature.com/articles/s41467-022-28882-9>
<https://www.sciencedirect.com/science/article/pii/S2211124722001528>

The HDX results presented do agree well with structural results obtained without the 2P mutations as described in these papers (which, by the way, are not cited):

[https://www.cell.com/molecular-cell/fulltext/S1097-2765\(22\)00266-0](https://www.cell.com/molecular-cell/fulltext/S1097-2765(22)00266-0)

[https://www.cell.com/cell-reports/fulltext/S2211-1247\(22\)00798-7](https://www.cell.com/cell-reports/fulltext/S2211-1247(22)00798-7)

Indeed, the agreement is quite striking not only with the results related to the higher proportion of the closed conformation, but also the changes in the S2 region that pre-dispose the Omicron spikes to undergo structural changes required for fusion. Bottomline, the differences observed between 2P and non-2P Omicron spikes, especially related to proportions of closed vs open conformations, are now well documented, and the results presented here, agree more with what was observed in non-2P spikes, although the spikes used for the HDX-MS experiments include the 2P mutations. This is an important discrepancy that must be addressed.

Other issues:

The title of figure 1 “Mechanism of transition from closed to open states” is misleading and over-reaching. This should be rephrased to indicate what the figure is showing, ie., Differences in HDX between Wuhan D614 and G614 spikes.

Line 150: “Hence, in all analyses we used a stabilized version of spike ectodomains containing ‘2P’ mutations, which make them furin-uncleavable and unable to transition to the postfusion conformation.” The 2P mutations do not have the spike furin uncleavable, the RRAR to GSAS substitution in the SD2 subdomain does.

In Figure 2, panel A, “D614 vs G614 (closed state)” not clear what the “closed state” here indicates.

Reviewer #2 (Remarks to the Author):

This manuscript describes H/D Exchange of the intact COVID spike protein, comparing all major variants of concern for shifts in conformational dynamics associated with 'open/closed' and 'fusion priming'. The paper is well written, the data are well presented and the results provide some important insights about mutation-dependent dynamic shifts in the NTD, Ace2 binding and fusion-priming. The evidence unambiguously supports the conclusions. I have only a few minor suggestions:

1. The Authors could do a better job of citing previous HDX studies on COVID spike, some of which provide highly relevant foundation and corroborating evidence for the current work. In particular, the work of Ganesh Anand should be cited. Also, the Wilson and Komives groups have measured dynamics of Spike by HDX in various contexts that may be relevant.
2. The 'bubble plots' are an excellent way of representing EX1 kinetics. It would be interesting to see these for more than one peptide in this region, perhaps in supplemental. To have full confidence in these data, similar kinetics for overlapping or nearby peptides should be included.
3. The idea of increased dynamics as a way of evading antibody binding is an interesting one. The authors may wish to reflect on how this mechanism would impact a continuous vs. discontinuous epitope in the discussion. (At first pass, my thought would be that this mode of evasion would be much more impactful on discontinuous epitopes than on continuous ones, and that it would never offer 'complete' protection, since the 'vulnerable' configuration would still occasionally be populated... Essentially, manipulation of conformer selection for antibody binding).

Reviewer #3 (Remarks to the Author):

The manuscript by Calvaresi reports an extensive HDX-MS comparative analysis of various CoV2 spike proteins in the unbound or ACE2-complexed states. The data are well presented, and the study can reveal some useful insight into the solution behavior of the spike protein. I applaud the authors for using freshly purified proteins to minimize confounding effects of freeze thaw that have previously been observed. The methods and the majority of the analysis is solid, including the direct comparison of the solution behavior of all the variants. There are a few major concerns going back to the experimental design for the ACE2 bound complexes as outlined below, along with a other minor issues listed below that need to be addressed.

Major concern:

Why did the authors use a ratio of 1:2 spike:ACE2? Won't this mean that 1 of the 3 RBDs of the spike is unbound? For the most conclusive HDX studies it is really important to favor full binding of your protein of interest so that you measure the full extent of changes associated with binding. With the 1:2 ratio used here, only a maximum of 66% of the spike RBD population can be in the bound form. This may manifest as two apparent populations for regions that are most perturbed by ACE2 binding.

Were bimodal spectra observed across different regions of the spike beyond the 495-513 region that was presented? Additionally, how long was the spike:ACE2 complex incubated prior to exchanges? Do

the authors know if this time was sufficient for equilibration to the 66% bound form?

Bimodal spectra at the 495-513 region presented look convincing. Based on the authors coverage maps each of the mutants have several peptides that span this same region in the spike proteins. The authors need to check and make sure that the bimodal (or peak broadening) is evident in all of those overlapping peptides too. I do not think it is critical for the authors to fit and thoroughly analyze all of the overlapping peptides, as it is not a central conclusion of the paper, but at the least the authors need to validate that it is observed consistently among the overlapping peptides and make a note of this in the results.

While the peak broadening is evident by eye, it also looks like the separation of the two populations within the bimodal is poor. Because of this, I do not think that there is high confidence associated with the fitting to extract the exact deuterium incorporation and exact sizes for the two populations. The authors interpret the bimodal as EX1 kinetics, but the two populations could easily result from the 66% that is ACE2 bound and the remaining 33% that is unbound. There will likely also be a distribution of different stoichiometries of ACE2-spike among the different spike molecules. For example there will be some finite number of spikes with all three lobes bound to ACE2, the majority with 2 lobes bound, a small fraction of only a single lobe bound, and a really small population of completely unbound spike. These different populations can easily result in complicated bimodal mass envelopes that have nothing to do with true EX1 kinetics. One thing the authors can do is to see if the higher deuterated population appears consistent with what was observed in the unbound spike. If bimodal profiles are observed in both the spike and the ACE2 then the authors might be able to interpret what fraction of the protein was actually bound.

These issues could have been alleviated if the experiment was designed with a sufficient excess of ACE2 to ensure fully bound complexes. I appreciate that these reagents are challenging to make and the experiments are difficult to carry out, and do not expect the authors to go back and redo the experiment. Furthermore, I think the data in this section can still be insightful, but the authors need to reexamine and rewrite this portion with full acknowledgment of all the confounding factors that limit how much can be concluded. The problem of incomplete ACE2 binding may also be a confounding factor for the following section on the S2 subunit priming.

Major point 2:

The authors examine HDX changes within ACE2 in complex with the various spike constructs to assess the binding affinity and avidity. This section is highly problematic. I am having trouble trying to understand the logic of why this experiment would reveal what the authors suggest. I agree that tighter binding will lead to more protection in the ACE2. Based on the methods the authors preincubated at a ratio of 1:2 spike trimer:ACE2. This ratio should correspond to a 3:2 ratio of spike monomer:ACE2, and if the interaction is of sufficient affinity then the vast majority of the ACE2 should be bound. For HDX studies you typically want an excess of the ligand with sufficient preincubation so you can be sure that you have near-full occupancy of the binding sites in the protein of interest. If not then you risk looking at a mixture of bound and unbound species that will depend on affinity and possibly incubation time. Depending on the association/dissociation kinetics this can also result in bimodal isotopic envelopes as the data might reflect a combination of free and bound populations, further complicating the analysis.

The authors suggest that since previous binding studies have indicate similar affinities between WT and G614, that a higher portion of the ACE2 is bound in the presence of G614 spikes. While this may be true, this would indicate that the complexes were not incubated long enough to reach equilibrium. At equilibrium with equal affinities there should be an identical amount of complex formed for both WT

and G614 spikes. If there isn't then it might relate to the formation of the ACE2-WT complex being much slower, perhaps because more time is needed for the WT to sample open conformations capable of binding.

Additionally, if the binding is incomplete then I would expect to see a bimodal isotopic profile near the binding site reflecting the population of bound and unbound ACE2. This bimodal may actually be a much more direct way to assess what portion of the ACE2 is able to bind the spike. However, in many cases the two populations in bimodal spectra are not well-resolved and quantifying the populations is difficult to determine with any confidence. Overall, this assay has a lot more variables and caveats than the authors account for, and I would be very hesitant to conclude anything from this data. I'm guessing the authors thought they could extract some additional information for the ACE2 since it's already in the dataset, but this is not a well-designed experiment for any solid conclusions. It might be a useful assay for future studies to examine kinetics of binding among the variants, but since this is not an integral part of the current paper, I advise the authors simply omit this section.

Minor point:

The authors should make a note that there is a caveat when comparing peptides among spike variants where the peptide sequences are not perfectly matched. The intrinsic rates of exchange will be offset by some of the amino acid substitutions, so it is possible to observe different exchange kinetics even if there is no actual difference in the structural dynamics of that region of the protein. The authors should make a note how they handled the comparison at regions where peptide sequences were not matched. It should help that the authors had a fully deuterated control so they can compare % deuteration instead of deuterium uptake, but this won't alleviate the caveat entirely. For example, one of the largest changes seen in with beta spikes was attributed K417N, and from the coverage map it looks like all of the peptides reporting around this region span residue 417. The authors should check how much of an effect the mutation will have on the intrinsic exchange rate. In many cases it will be relatively minor and the magnitude of the observed difference in exchange will surpass any effect of the point mutation. However, in some cases, for example proline mutations, the number of amides will be affected and therefore have a larger effect.

This paper has a description of how peptides with divergent sequences can be handled for comparative HDX studies: doi: 10.1007/s13361-016-1365-5

Reviewer #1 (Remarks to the Author):

In the manuscript titled “Structural dynamics in the evolution of 1 SARS-CoV-2 spike glycoprotein”, Calvaresi et al. utilize HDX-MS to study the structure and dynamics of the SARS-CoV-2 spike protein. They relate their observations to published structures of the spike.

Several of the results presented are consistent with expectations based on existing structural data and/or add to our understanding of the SARS-CoV-2 spike structure and dynamics. These include the increase in HDX in regions of the spike where increased flexibility/mobility is expected because of the D614G mutation, the decreased HDX in the regions spanning the ACE2-RBD interface, ACE2-induced changes in the S2 subunit, and the observation of increased inner flexibility of the Omicron spike RBD that agrees with observations of differing thermostability of the Omicron RBDs ([https://www.cell.com/cell-reports/fulltext/S2211-1247\(22\)00798-7](https://www.cell.com/cell-reports/fulltext/S2211-1247(22)00798-7)).

There are other observations, however, that do not agree with structural data despite the authors’ claims that they do. Notable among these is the observation, which is also a primary result/conclusion of this study that the “substitutions in omicron spike lead to predominantly closed conformations, presumably enabling it to escape antibodies”. The constructs studied in this paper include the 2P mutations. Published cryo-EM structures of Omicron spikes that include the 2P mutations show a preponderance of open states. Two examples of these are:

<https://www.nature.com/articles/s41467-022-28882-9>

<https://www.sciencedirect.com/science/article/pii/S2211124722001528>

The HDX results presented do agree well with structural results obtained without the 2P mutations as described in these papers (which, by the way, are not cited):

[https://www.cell.com/molecular-cell/fulltext/S1097-2765\(22\)00266-0](https://www.cell.com/molecular-cell/fulltext/S1097-2765(22)00266-0)

[https://www.cell.com/cell-reports/fulltext/S2211-1247\(22\)00798-7](https://www.cell.com/cell-reports/fulltext/S2211-1247(22)00798-7)

Indeed, the agreement is quite striking not only with the results related to the higher proportion of the closed conformation, but also the changes in the S2 region that pre-dispose the Omicron spikes to undergo structural changes required for fusion. Bottomline, the differences observed between 2P and non-2P Omicron spikes, especially related to proportions of closed vs open conformations, are now well documented, and the results presented here, agree more with what was observed in non-2P spikes, although the spikes used for the HDX-MS experiments include the 2P mutations. This is an important discrepancy that must be addressed.

We thank the reviewer for pointing out this discrepancy, as indeed, in the interest of saving space, we oversimplified this issue. Following the reviewer’s suggestion, we have now addressed it in the lines 435-441 and 445-446 of the discussion session.

Other issues:

The title of figure 1 “Mechanism of transition from closed to open states” is misleading and over-

reaching. This should be rephrased to indicate what the figure is showing, i.e., Differences in HDX between Wuhan D614 and G614 spikes.

We thank the reviewer for the suggestion. We have now changed the figure caption in 'Differences in structural dynamics between Wuhan (D614) spike and G614 spike'

Line 150: "Hence, in all analyses we used a stabilized version of spike ectodomains containing '2P' mutations, which make them furin-uncleavable and unable to transition to the postfusion conformation." The 2P mutations do not have the spike furin uncleavable, the RRAR to GSAS substitution in the SD2 subdomain does.

We now changed this sentence to: *'Hence, in all analyses we used a version of spike ectodomains stabilized by R682S, R685S substitutions in the polybasic cleavage sites and K986P, K987P (2P) substitutions, which combined make the spikes furin-uncleavable and unable to transition to the postfusion conformation.'* (lines 143-147)

In Figure 2, panel A, "D614 vs G614 (closed state)" not clear what the "closed state" here indicates.

We have now removed a reference to the 'closed state' in the figure.

Reviewer #2 (Remarks to the Author):

This manuscript describes H/D Exchange of the intact COVID spike protein, comparing all major variants of concern for shifts in conformational dynamics associated with 'open/closed' and 'fusion priming'. The paper is well written, the data are well presented and the results provide some important insights about mutation-dependent dynamic shifts in the NTD, Ace2 binding and fusion-priming. The evidence unambiguously supports the conclusions. I have only a few minor suggestions:

1. The Authors could do a better job of citing previous HDX studies on COVID spike, some of which provide highly relevant foundation and corroborating evidence for the current work. In particular, the work of Ganesh Anand should be cited. Also, the Wilson and Komives groups have measured dynamics of Spike by HDX in various contexts that may be relevant.

We thank the reviewer for the suggestions. We had cited two of Ganesh Anand's papers (ref. 35 and 39), and we have now included his recent preprint (ref. 40). We have also now included the work for Wilson group on the RBD (ref. 41) and a very recent article based on mass photometry data (ref. 42). However, we could not find any HDX-MS study on spike conducted by Komives group.

2. The 'bubble plots' are an excellent way of representing EX1 kinetics. It would be interesting to see these for more than one peptide in this region, perhaps in supplemental. To have full confidence in these data, similar kinetics for overlapping or nearby peptides should be included.

We thank the reviewer for this suggestion. We have now analysed with HX-Express2 another peptide spanning the RBM (peptide 495-510) and included this extra analysis in the supplementary information

(a peptide spanning the same residues is unfortunately absent in the omicron spike) (Fig. S44-45). It is worth noting that the HX-express fitting returned very similar relative size of populations and their time-dependent evolution, for every spike variant, as the peptide previously included. We have also extracted from DynamX the stacked spectral plots of many other peptides encompassing this region, and included them in the supplementary figures (Fig. S41). We included peptides of both high and moderate data quality, to demonstrate that bimodal spectra, or peak broadening, were visually distinguishable across all of them, regardless the peptide signal-to-noise ratio. Here below the additional figures.

Supplementary Fig. 41 (a,b). EX1 kinetics in selected peptides spanning residues 495-503 of spike receptor binding motif (RBM) of spike trimers and the isolated RBD in complex with ACE2. Stacked spectral plots elucidate the evolution of the bimodal isotopic envelopes across the time points studied and for different overlapping peptides in spike trimers. The bimodal isotopic envelopes of peptides 495-510 and 495-513 have been also analysed with HX-Express² (see Supplementary Figs. 42-45). The isotopic envelopes of the isolated ancestral RBD does not display bimodal features nor evident peak broadening.

Supplementary Fig. 44. EX1 kinetics in the receptor binding motif (RBM) of spikes in complex with ACE2. From top to bottom: peptide YGFQPTNGVGYQP YRV (495-510) of Wuhan spike; peptide YGFQPTNGVGYQP YRV (495-510) of G614 spike; peptide YGFQPTNGVGYQP YRV (493-509) of delta spike; peptide YGFQPTYGVGYQP YRV (492-507) of alpha spike; peptide YGFQPTYGVGYQP YRV (492-507) of beta spike. From left to right: bubble plots representing the level of deuteration and the relative intensity of the low- and high-mass envelope (binomial fit 1 and binomial fit 2, respectively) for bimodal isotopic distributions and the level of deuteration of the unimodal envelope (unimodal fit), set at 100% intensity, in the ACE2 bound state; peak width of the isotopic distribution (calculated at 20% of Baseline Peak Intensity -BPI) and number of exchangeable amides (#NHs), in the bound state; deuterium level of the low-mass envelopes (binomial 1), high-mass envelopes (binomial 2) and unimodal distributions (centroid) in ACE2-bound spike; deuterium level of spikes in apo state (unimodal distributions – centroid). Spectral analysis was conducted with HX-Express2² and graphs were automatically generated upon fitting the isotopic envelopes shown in Supplementary Fig. 45. Bimodal fit was applied to peptide spectra at time points showing clear features of bimodality and/or enhanced peak width. Note that the high-mass population does not manifest at early time points, thus does not follow the HDX of the unbound state.

a

b

Supplementary Fig. 45. EX1 kinetics in the receptor binding motif (RBM) of spikes in complex with ACE2. Mass spectra of the representative peptides of the RBM (listed in Supplementary Fig. 44) deconvoluted with bimodal or unimodal fittings with HX-Express 2. A) Mass spectra of peptide of Wuhan, G614 and delta spikes. B) Mass spectra of peptide of alpha, beta and omicron spikes. The horizontal orange lines indicate the distribution width at 20% BPI; the red dots mark the envelope peaks; the vertical green lines indicate the centroid mass; the dark blue lines fit the unimodal envelope distributions deconvoluted with binomial fitting; the green lines fit the low-mass envelope distributions deconvoluted with bimodal fitting; the light blue lines fit the high-mass envelope distributions deconvoluted with bimodal fitting.

3. The idea of increased dynamics as a way of evading antibody binding is an interesting one. The authors may wish to reflect on how this mechanism would impact a continuous vs. discontinuous epitope in the discussion. (At first pass, my thought would be that this mode of evasion would be much more impactful on discontinuous epitopes than on continuous ones, and that it would never offer 'complete' protection, since the 'vulnerable' configuration would still occasionally be populated... Essentially, manipulation of conformer selection for antibody binding).

This is a very interesting input and we completely align with the reviewer's thoughts. We have added the following paragraph in the discussion (lines 410-415):

"The conformational plasticity of this NTD site presumably allows antibody-vulnerable configurations of conformational epitopes to remain occasionally populated, reducing, but not completely abrogating, antibody-mediated protection. At the same time, linear NTD epitopes are likely more impacted by the residue changes in spike variants rather than by a different conformational aspect of the NTD of VOCs."

Reviewer #3 (Remarks to the Author):

The manuscript by Calvaresi reports an extensive HDX-MS comparative analysis of various CoV2 spike proteins in the unbound or ACE2-complexed states. The data are well presented, and the study can reveal some useful insight into the solution behavior of the spike protein. I applaud the authors for using freshly purified proteins to minimize confounding effects of freeze thaw that have previously been observed. The methods and the majority of the analysis is solid, including the direct comparison of the solution behavior of all the variants. There are a few major concerns going back to the experimental design for the ACE2 bound complexes as outlined below, along with a other minor issues listed below that need to be addressed.

Major concern:

Why did the authors use a ratio of 1:2 spike:ACE2? Won't this mean that 1 of the 3 RBDs of the spike is unbound? For the most conclusive HDX studies it is really important to favor full binding of your protein of interest so that you measure the full extent of changes associated with binding. With the 1:2 ratio used here, only a maximum of 66% of the spike RBD population can be in the bound form. This may manifest as two apparent populations for regions that are most perturbed by ACE2 binding.

This an important point and we thank the reviewer for the opportunity to clarify it. For a Spike trimer to be ACE2 binding-competent at least one of its RBDs must be erect and a mixture of its open and closed states within the trimer is well known in the literature (see e.g. <https://doi.org/10.1016/j.cell.2020.02.058>, <https://www.nature.com/articles/s41594-020-0468-7.pdf?origin=ppub>, <https://www.pnas.org/doi/10.1073/pnas.2022586118>, and <https://www.nature.com/articles/s41467-022-28882-9>). Spikes with all three RBDs accessible are rarely observed, at least in cryoEM. Although we cannot be sure of the precise ratios of up and down states in spikes in solution, in non-cryogenic state, the conformational dynamics between these adopted structures governs this interaction and is what our solution HDX-MS experiments can report on. In our experiments we, therefore, try to capture a situation where we can monitor this behaviour within the experimental limitations of required sample amounts and concentration – with large concentrations of the material difficult to produce.

The 1:2 ratio was chosen after initial optimization HDX-MS experiments. We incubated Wuhan and G614 spikes at 1:2 and 1:3 ratio spike:ACE2 and found the same difference in HDX for spike peptides within the RBM at both ratios. This suggests that increased ACE2 above 1:2 was not increasing spike binding occupancy, but likely leads to an increased population of unbound ACE2 in our labelled sample. Accordingly, the HDX differences observed within ACE2 incubated at 1:2 and 1:3 ratios spike:ACE2 appeared significantly higher at ratio 1:2 than at 1:3, which supports that a higher fraction of unbound ACE2 is present in the latter case. In support of the ratio chosen, a very recent mass photometry paper (<https://pubs.rsc.org/en/content/articlelanding/2022/cc/d2cc04711j>) on Wuhan and omicron spikes (in the same '2P' pre-fusion stabilised version we use) reported that 1:4 spike:ACE2 shows predominantly spike bound to one ACE2, with some spike bound to 2 ACE2 but negligible amounts bound to 3 ACE2.

Considering these observations, 1:2 ratio likely provides information on saturated spike ACE2-accessible 'up' states. As the reviewer points out, this would mean that one of the RBDs will be unbound, but it is likely to be in an ACE2-inaccessible "down" state. The experiment performed to address major concern 2 provides further evidences in support of the ratio chosen, for every spike analysed (please see below).

In this revised version, we inserted this optimization procedure in a dedicated paragraph of the method session (*Optimization of the HDX conditions – lines 546-554*) and added two figures showing these results in the supplementary information and present them below.

Fig. S47. Histogram illustrating the magnitude of decreased HDX (Δ HDX) manifesting in selected peptides of the RBM when Wuhan and G614 spikes are incubated at 1:2 and 1:3 ratios with ACE2. No significant differences in Δ HDX are observed between the two incubation ratios and for both spike trimers.

Supplementary Fig. 48. Histogram illustrating the magnitude of decreased HDX (Δ HDX) manifesting in selected peptides of the ACE2 binding site when ACE2 is incubated at 2:1 and 3:1 ratios with Wuhan spike. A minor decrease in HDX is observed at ratio 3:1.

Were bimodal spectra observed across different regions of the spike beyond the 495-513 region that was presented?

Bimodal spectra were not observed in apo spikes by working with freshly prepared spike samples. In the ACE2-bound state, beyond the 495-513 region, we observed - for every spike trimer analyzed - the appearance of bimodal isotopic envelopes in the HR1 region spanning residues 962-982, which is one of the segments becoming more dynamic during the priming for fusion of spike. In the interest of writing the manuscript for a broad readership, we did not describe the specific HDX regime of this particular segment in detail, rather summarizing the increased HDX observed in various S2 stretches as 'increased dynamics'. However, following the reviewer's inquiry, we have reconsidered our initial thoughts and have now described the HDX bimodality seen in this region of bound-spikes in the manuscript text, as presumably it also contributes to the priming mechanism (lines 355-363).

"Notably, peptides spanning residues 962-982 manifested increased HDX in the form of bimodal isotopic envelopes when spikes are engaged to ACE2 (Fig. 6c and Supplementary Fig. 46). This segment encompasses the HR1 helix, which undergoes a large conformational rearrangement upon the transition to post-fusion state¹⁵, suggesting that our analysis captured the specific dynamic events leading to the HR1 reorientation, which primes spikes for fusion. The breadth of this HDX bimodality varies with omicron > alpha > beta > G614 ≈ Wuhan spike, with omicron spike pre-manifesting it also in the absence of ACE2 (Fig. 6c and Supplementary Fig. 46)."

Additionally, we have modified Fig. 6c in the main text to include the isotopic envelopes of peptides 963-977, in place of its uptake plots. We believe that the envelopes deliver a much clearer message. We also added a supplementary figure (S46) including the stacked spectral plots of peptide 963-977 and an overlapping peptide (962-977).

Fig. 6 Spike priming for fusion. A) Regions manifesting increased dynamics in spikes bound to ACE2 are superimposed and colored in red on a single protomer of the structure of D614 spike with one RBD bound (PDB: 7a95¹⁴); regions with increased dynamics only in spike of alpha and omicron variants bound to ACE2 are colored in magenta. B) The magnitude of the destabilization of the core helices is represented by differential colouring (red scale) for the various spike trimers. The HR1 region (962-982) manifesting HDX bimodality is framed in grey. C) The bimodal isotopic envelopes of a model peptide spanning the HR1 region 962-982 are shown at 15 s and 1 min time points for spike (S) variants in the ACE2-bound form and for omicron spike alone, to exemplify their priming for fusion upon receptor engagement. Bimodal envelopes manifest in omicron spike also in the absence of ACE2, indicating it as pre-primed for fusion.

Supplementary Fig. 46. HDX bimodality in the HR1 of spikes in complex with ACE2 and omicron spike alone. Stacked spectral plots of peptides 962-977 and 963-977 illustrate the evolution of the bimodal isotopic envelopes over the time points studied. Particularly, time points 15 s and 1 min (23 °C) show clear separation between the low- and high-mass envelopes in every spike trimer, with the relative intensity and centroids of the two envelopes reporting on the degree of destabilization exerted by the ACE2 binding on the HR1. The isotopic envelopes in omicron spike alone manifest evident peak broadening, suggesting that its HR1 follows a bimodal HDX behaviour even in the absence of ACE2.

Additionally, how long was the spike:ACE2 complex incubated prior to exchanges? Do the authors know if this time was sufficient for equilibration to the 66% bound form?

Spikes-ACE2 complexes were incubated for one hour before starting the deuterium labelling (we now specified this in the method session). Given the Kds are, depending on the variant, in the order of 10 to 150 nM with kons between 0.07 and 0.2 $\mu\text{M}^{-1}\text{s}^{-1}$ (<https://doi.org/10.1038/s41467-022-28768-w> - see supplementary info), we believe that this incubation period largely suffices for system equilibration.

Bimodal spectra at the 495-513 region presented look convincing. Based on the authors' coverage maps each of the mutants have several peptides that span this same region in the spike proteins. The authors need to check and make sure that the bimodal (or peak broadening) is evident in all of those overlapping peptides too. I do not think it is critical for the authors to fit and thoroughly analyze all of the overlapping peptides, as it is not a central conclusion of the paper, but at the least the authors need to validate that it is observed consistently among the overlapping peptides and make a note of this in the results.

We thank the reviewer for highlighting this important point. We have now extracted from DynamX the stacked spectral plots of many other peptides encompassing this region and included them in the supplementary figures. We included peptides of both high and moderate data quality, to demonstrate that bimodal spectra, or peak broadening, were visually distinguishable across all of them, regardless of the peptide signal-to-noise ratio (Fig. S41). We also analysed the bimodal isotopic distributions of another peptide (495-510) with HX-express2 and included this extra analysis in the supplementary information (a matching peptide is unfortunately absent in the omicron variant) (Fig. S44-45). It is worth noting that the HX-express2 fitting returned very similar relative size of populations and their time-dependent evolution, for every spike variant, compared to the peptide previously included. The result of this extra analysis could also address the reviewer's following concern about the calculated size of the populations.

Supplementary Fig. 41 (a,b). EX1 kinetics in selected peptides spanning residues 495-503 of spike receptor binding motif (RBM) of spike trimers and the isolated RBD in complex with ACE2. Stacked spectral plots elucidate the evolution of the bimodal isotopic envelopes across the time points studied and for different overlapping peptides in spike trimers. The bimodal isotopic envelopes of peptides 495-510 and 495-513 have been also analysed with HX-Express² (see Supplementary Figs. 42-S45). The isotopic envelopes of the isolated ancestral RBD does not display bimodal features nor evident peak broadening.

Supplementary Fig. 44. EX1 kinetics in the receptor binding motif (RBM) of spikes in complex with ACE2. From top to bottom: peptide YGFQPTNGVGYQPYRV (495-510) of Wuhan spike; peptide YGFQPTNGVGYQPYRV (495-510) of G614 spike; peptide YGFQPTNGVGYQPYRV (493-509) of delta spike; peptide YGFQPTYG VGYQPYRV (492-507) of alpha spike; peptide YGFQPTYG VGYQPYRV (492-507) of beta spike. From left to right: bubble plots representing the level of deuteration and the relative intensity of the low- and high-mass envelope (binomial fit 1 and binomial fit 2, respectively) for bimodal isotopic distributions and the level of deuteration of the unimodal envelope (unimodal fit), set at 100% intensity, in the ACE2 bound state; peak width of the isotopic distribution (calculated at 20% of Baseline Peak Intensity -BPI) and number of exchangeable amides (#NHs), in the bound state; deuterium level of the low-mass envelopes (binomial 1), high-mass envelopes (binomial 2) and unimodal distributions (centroid) in ACE2-bound spike; deuterium level of spikes in apo state (unimodal distributions – centroid). Spectral analysis was conducted with HX-Express2² and graphs were automatically generated upon fitting the isotopic envelopes shown in Supplementary Fig. S45. Bimodal fit was applied to peptide spectra at time points showing clear features of bimodality and/or enhanced peak width. Note that the high-mass population does not manifest at early time points, thus does not follow the HDX of the unbound state.

a

b

Supplementary Fig. 45. EX1 kinetics in the receptor binding motif (RBM) of spikes in complex with ACE2. Mass spectra of the representative peptides of the RBM (listed in Supplementary Fig. 44) deconvoluted with bimodal or unimodal fittings with HX-Express 2. a) Mass spectra of peptide of Wuhan, G614 and delta spikes. b) Mass spectra of peptide of alpha, beta and omicron spikes. The horizontal orange lines indicate the distribution width at 20% BPI; the red dots mark the envelope peaks; the vertical green lines indicate the centroid mass; the dark blue lines fit the unimodal envelope distributions deconvoluted with binomial fitting; the green lines fit the low-mass envelope distributions deconvoluted with bimodal fitting; the light blue lines fit the high-mass envelope distributions deconvoluted with bimodal fitting.

While the peak broadening is evident by eye, it also looks like the separation of the two populations within the bimodal is poor. Because of this, I do not think that there is high confidence associated with the fitting to extract the exact deuterium incorporation and exact sizes for the two populations. The authors interpret the bimodal as EX1 kinetics, but the two populations could easily result from the 66% that is ACE2 bound and the remaining 33% that is unbound. There will likely also be a distribution of different stoichiometries of ACE2-spike among the different spike molecules. For example there will be some finite number of spikes with all three lobes bound to ACE2, the majority with 2 lobes bound, a small fraction of only a single lobe bound, and a really small population of completely unbound spike. These different populations can easily result in complicated bimodal mass envelopes that have nothing to do with true EX1 kinetics. One thing the authors can do is to see if the higher deuterated population appears consistent with what was observed in the unbound spike. If bimodal profiles are observed in both the spike and the ACE2 then the authors might be able to interpret what fraction of the protein was actually bound.

These issues could have been alleviated if the experiment was designed with a sufficient excess of ACE2 to ensure fully bound complexes. I appreciate that these reagents are challenging to make and the experiments are difficult to carry out, and do not expect the authors to go back and redo the experiment. Furthermore, I think the data in this section can still be insightful, but the authors need to reexamine and rewrite this portion with full acknowledgment of all the confounding factors that limit how much can be concluded. The problem of incomplete ACE2 binding may also be a confounding factor for the following section on the S2 subunit priming.

We thank the reviewer for considering this important aspect. We based our interpretation on the following observations and experiments.

As the reviewer suggests, we had carefully compared the HDX of the high-mass population of the bound states and the HDX of the population (unimodal spectra) of the apo state (also shown in fig. S42 and S44). The two HDX profiles do not appear consistent. We believe that, if the ACE2 was not sufficient to saturate the binding of the monomers with accessible (erect) RBD, the high-mass population in the bound states should align on the m/z scale to the apo spike states, manifesting already, and with clear separation from the low-mass population, at 4 s or 20 s on ice, but this does not appear to be the case. We do not observe it at 4 s on ice for Wuhan and G614 bound-spikes, or at 20 s on ice for alpha, beta and omicron bound-spikes. We have now added a note on this in the figure caption of fig. S42 and S44.

To better investigate this aspect, for this revised version, we performed an extra experiment. We performed HDX-MS on the isolated RBD of the ancestral Wuhan spike in the presence and absence of ACE2, at ratio 3:2 RBD:ACE2, which simulates our 1:2 spike trimer:ACE2 ratio used in our experiments. Given that the whole population of isolated RBD molecules are binding-competent, with such a binding stoichiometry, 33% of the RBD population remain effectively unbound in the presence of ACE2. We did not observe bimodal distributions in the RBM of the bound-RBD state, indicating that a mixture of bound and unbound monomers, with same conformational characteristics in the apo state, does not manifest with a clear envelope bimodality under the conditions studied. We now included the extracted stacked spectral plot of the bound-RBD in the supplementary figures (please, see previous figure S41). Furthermore, the preliminary data shown above and the experiment performed to address major point 2 support a scenario where spike erect RBDs are saturated in binding occupancy.

These observations prompt us to associate the RBM bimodal HDX profiles to the spike cooperative binding mode reported in previous studies; we thus interpreted the high-mass population as an extra monomer that erects upon ACE2 binding (in a cooperative manner) and is likely able to engage with an extra ACE2 molecule, thus displaying an HDX profile that differs from the monomers in the unbound state. This gives rise to a fine-tuned equilibrium of states, characteristic for each spike variant, that we aimed at deciphering with our analysis.

We agree with the reviewer that we should not strictly refer to this HDX behaviour as EX1 kinetics, as we associated it to the behaviour of different monomers, and not to an individual protein stretch displaying correlated exchange. We thus now refer to it as “HDX bimodality”. We also modified the text (lines 306-324) to better explain these observations and include the extra experiment performed, and we present it here below.

“Furthermore, we observed that the HDX profiles of all peptides spanning the RBM of spike trimers (residues 495-503) in the ACE2-bound states showed bimodal isotopic distributions, hence a high- and a low-mass population, whereas a single unimodal distribution characterized the apo states (Fig. 5 and Supplementary Figs. 41-45). The HDX of the high-mass populations in the bound states appeared inconsistent with the HDX of the respective apo states. In contrast, the ACE2-bound state of the isolated ancestral RBD (3:2 RBD: ACE2), containing a significant fraction (33%) of unbound population, did not display bimodal isotopic distributions in the RBM (Supplementary Fig. 41), ruling out that a mixture of bound and unbound populations, with same conformational characteristics in the apo state, manifests with an HDX bimodality under the conditions studied. These data thus suggest that the RBD of bound spikes can explore two distinct and slowly interconverting populations, which exchange giving rise to two resolved isotopic distributions. We rationalize, based on the receptor binding mode reported in previous studies^{14,55}, that the bimodal HDX profile of the bound spike states reports on cooperative opening within the spike trimer, with the less exchanged (low-mass) population accounting for open protomers with a bound RBD, whilst the more exchanged (high-mass) population likely corresponding to closed protomers transitioning to the open state and readying to engage another receptor molecule.”

We also would like to highlight that bimodal spectra were not observed in the bound states of ACE2, even though we observed these states at uncomplete binding occupancy (please, see answer to Major point 2).

Major point 2:

The authors examine HDX changes within ACE2 in complex with the various spike constructs to assess the binding affinity and avidity. This section is highly problematic. I am having trouble trying to understand the logic of why this experiment would reveal what the authors suggest. I agree that tighter binding will lead to more protection in the ACE2. Based on the methods the authors preincubated at a ratio of 1:2 spike trimer:ACE2. This ratio should correspond to a 3:2 ratio of spike monomer:ACE2, and if the interaction is of sufficient affinity then the vast majority of the ACE2 should be bound. For HDX studies you typically want an excess of the ligand with sufficient preincubation so you can be sure that you have near-full occupancy of the binding sites in the protein of interest. If not then you risk looking at a mixture of bound and unbound species that will depend on affinity and possibly incubation time. Depending on the association/dissociation kinetics this can also result in bimodal isotopic envelopes as the data might reflect a combination of free and bound populations, further complicating the analysis.

The authors suggest that since previous binding studies have indicated similar affinities between WT and G614, that a higher portion of the ACE2 is bound in the presence of G614 spikes. While this may be true, this would indicate that the complexes were not incubated long enough to reach equilibrium. At equilibrium with equal affinities there should be an identical amount of complex formed for both WT and G614 spikes. If there isn't then it might relate to the formation of the ACE2-WT complex being much slower, perhaps because more time is needed for the WT to sample open conformations capable of binding.

The reviewer is right; however, the experiments we referred to were all based on methods (biolayer interferometry) that measure only the affinity of a single monomer binding to one ACE2, as spikes are immobilised and thus their concentrations (as well as the effective concentrations of individual accessible RBDs) irrelevant for the measurement, while ACE2 monomers form a mobile phase. We have now specified this in the text (lines 269-270). In our HDX-MS experiments, we equilibrated all complexes for one hour before labelling, which, considering the aforementioned favourable k_{on} and K_d , should allow for equilibration to be reached.

Additionally, if the binding is incomplete then I would expect to see a bimodal isotopic profile near the binding site reflecting the population of bound and unbound ACE2. This bimodal may actually be a much more direct way to assess what portion of the ACE2 is able to bind the spike. However, in many cases the two populations in bimodal spectra are not well-resolved and quantifying the populations is difficult to determine with any confidence. Overall, this assay has a lot more variables and caveats than the authors account for, and I would be very hesitant to conclude anything from this data. I'm guessing the authors thought they could extract some additional information for the ACE2 since it's already in the dataset, but this is not a well-designed experiment for any solid conclusions. It might be a useful assay for future studies to examine kinetics of binding among the variants, but since this is not an integral part of the current paper, I advise the authors simply omit

We thank the reviewer for highlighting this important point and below we provide further details to support our experimental design and their conclusions.

Despite the molar excess of spike monomers, ACE2 is in excess compared to the binding-competent open monomers with available erect RBDs within spike trimers. To confirm this also in HDX-MS, we performed an additional experiment for this revised version of the manuscript. We labelled ACE2 in the presence and absence of the isolated ancestral RBD, at a 3:2 ratio RBD:ACE2, which simulates a 1:2 spike trimer:ACE2 ratio. Differently from the RBDs embedded in a spike trimer, the whole population of isolated RBD is binding-competent. In this scenario, the RBD is effectively in excess compared to ACE2, granting full ACE2 binding occupancy, and indeed, we observed that the binding effect on ACE2 manifested with much greater magnitude (cumulative $\Delta\text{HDX} = 19.15$ Da) compared to that induced by spike trimers, which show generally less cumulative ΔHDX due to the presence of a (varied) fraction of unbound ACE2 in the bound state. This experiment indicates that the ACE2 is not saturated in binding occupancy in the HDX-MS experiments performed on spike trimers (hence spike trimers are saturated or close to saturated), therefore we argue that the magnitude of HDX effects (ΔHDX) on the ACE2 holds useful information on the spike-ACE2 binding stoichiometry.

It has to be noted that the ΔHDX induced by alpha, beta and delta spikes also have a contribution from the stability of the hydrogen bonding network engaged with ACE2 (related to the affinity of their individual binding-competent monomers to the receptor), which will result in a cumulative effect given by their binding stoichiometry and affinity (reported as higher than for G614 in several studies). This is presumably the reason why for alpha spike a cumulative ΔHDX slightly higher than that of the isolated RBD was observed.

In our hands, the incomplete binding occupancy of ACE2 did not manifest with isotopic bimodal distributions in its spike binding sites, making it difficult to estimate the fraction of unbound ACE2 for the different trimers. This is one of the reasons why we refrain from proposing any stoichiometry model.

We have now included the results of this extra experiment (see figures below) in fig. 3 of the main text and Fig S27 and S40 of supplementary information and have also rewritten this section in the main text to provide a clearer explanation of our rationale (lines 243-255).

“Next, by studying the HDX of the ACE2 ectodomain alone and in complex with spike trimers (1:2 spike trimer:ACE2) and the isolated ancestral RBD (3:2 RBD:ACE2), we measured the magnitude of the HDX effects (ΔHDX) induced by spike binding to ACE2. The whole population of the isolated RBD is binding competent, granting complete occupancy of the ACE2 binding sites, whereas only a fraction of the RBDs embedded within spikes are erect and thus able to engage the receptor. The observed ΔHDX results from a cumulative effect of binding stoichiometry (how many ACE2 molecules are bound) and the stability of the hydrogen-bonding network between spikes and ACE2 (which can be related to the spike-receptor binding affinity), enabling us to rank the spike-receptor binding avidity (i.e. the overall strength of binding arising from the affinity of an individual RBD-ACE2 interaction and the stoichiometry of each spike trimer engaging between zero and three ACE2 molecules at once).”

For clarity, we also modified the following sentence (lines 261-268):

“The cumulative ΔHDX induced by different spikes varied with $\alpha > \beta > \delta > \text{G614} > \text{Wuhan} \approx \text{omicron}$ (Fig. 3b and Supplementary Fig. 28). These ΔHDX values were generally lower than that induced by the isolated RBD, indicating that a fraction of ACE2 molecules remained unbound in the spike:ACE2 states, thus suggesting that all binding-competent RBDs within the trimers were fully occupied.”

Fig. 3 Effect of spike binding on ACE2 dynamics. A) Regions of ACE2 manifesting a significant decrease in HDX upon spike binding are superimposed on the structure of ACE2 ectodomain bound to RBD (PDB: 2ajf), colored in blue scale according to the magnitude of the HDX effect. The region colored in red indicates increased HDX upon binding, in dark gray regions with no coverage. B) ACE2 binding avidity. The cumulative difference in HDX (Δ HDX) between ACE2 alone and ACE2 bound to spikes and the isolated ancestral RBD for selected peptides spanning binding sites and across time points 20 s on ice, 10 min at 23 °C and 360 min at 28 °C is plotted. A plot for all time points in Supplementary Fig. 28.

Supplementary Fig 27. Difference plot illustrating the difference in HDX between ACE2 in complex with the isolated ancestral RBD and ACE2 alone (orange line indicates: 20 s on ice, green line: 10 min at 23 °C, dark blue line: 360 min at 28 °C). Peptide segments of interest are highlighted. The peptides are arranged according to their peptide centre residue. A dotted grey line indicates the 98% CI as a threshold for significance and a dotted black line the 99% CI as a threshold for significance.

Supplementary Fig 40. Difference plot illustrating the difference in HDX between the isolated ancestral RBD in complex with ACE2 and the isolated ancestral RBD alone (orange line: 20 s on ice, green line: 10 min at 23 °C, dark blue line: 360 min at 28 °C). Residues comprising a region with significant differences in HDX are indicated. The peptides are arranged according to their peptide centre residue. A dotted grey line indicates the 98% CI as a threshold for significance.

Minor point:

The authors should make a note that there is a caveat when comparing peptides among spike variants where the peptide sequences are not perfectly matched. The intrinsic rates of exchange will be offset by some of the amino acid substitutions, so it is possible to observe different exchange kinetics even if there is no actual difference in the structural dynamics of that region of the protein. The authors should make a note how they handled the comparison at regions where peptide sequences were not matched. It should help that the authors had a fully deuterated control so they can compare % deuteration instead of deuterium uptake, but this won't alleviate the caveat entirely. For example, one of the largest changes seen in with beta spikes was attributed K417N, and from the coverage map it looks like all of the peptides reporting around this region span residue 417. The authors should check how much of an effect the mutation will have on the intrinsic exchange rate. In many cases it will be relatively minor and the magnitude of the observed difference in exchange will surpass any effect of the point mutation. However, in some cases, for example proline mutations, the number of amides will be affected and therefore have a larger effect.

This paper has a description of how peptides with divergent sequences can be handled for comparative HDX studies: doi: 10.1007/s13361-016-1365-5

We gratefully thank the reviewer for highlighting the caveat arising from the comparison of peptides harboring mutations and suggesting the paper (which we now cited). We briefly explained in the method section how they were compared but did not consider this to a sufficient degree. We explain our comparative workflow below and the additional k_{ch} considerations done:

Workflow:

We selected for comparison only peptides with identical cleavage, i.e. same N- and C-termini. Thanks to the numerous peptides available and the high redundancy, we could, in most instances, find matching peptides. However, this became impossible in case of deletions and insertions. We normalized the uptake values of mutant peptides by the uptake of their MaxD (= fully deuterated control) and obtained absolute uptake values (in Da) referencing to G614 spike, with the following equation (now included in the method session):

$$\Delta HDX = \left(\frac{DU \text{ mutant peptide}}{DU \text{ MaxD mutant peptide}} \times DU \text{ MaxD G614 spike peptide} \right) - DU \text{ G614 spike peptide}$$

For instance, at a given time point, peptide X of G614 spike has DU of 3 Da with a MaxD of 10 Da; its matching peptide Y in alpha spike has DU of 4 Da with MaxD of 10.3 Da; the normalized DU for alpha spike is 3.88 Da. The ΔHDX reported in the butterfly plot is +0.88 Da. This was done with the aim to insert mutant peptides in the butterfly plots, making the HDX comparison more visually intuitive and easily readable.

Considering k_{ch} differences:

We acknowledge the fact that the MaxD does not entirely alleviate the caveat arising from the difference in k_{ch} between peptides harboring mutations. Before performing the experiments, we had checked the values of k_{ch} (according to the 2018 updated values: doi: 10.1007/s13361-018-2021-z) and

noticed that the differences are minor and most likely give rise to ΔHDX below the threshold of significance when comparing peptides without a significant difference in k_{op} .

However, we now performed a more thorough analysis on the impact that the k_{ch} offset has on the differences measured. We included this analysis in a supplementary table (Supplementary Data 1). We tested a null hypothesis in which every difference in HDX observed arises from a difference in k_{ch} .

In more detail: we calculated the average k_{ch} for a peptide and its mutant variant - excluding the N-terminal residue (one peptide per mutation was analyzed). The average k_{ch} was selected as the individual amide HDX rates are averaged when measured by MS at peptide level. We then calculated the % of observed ΔHDX for that peptide normalized by MaxD, selecting the time point showing the highest ΔHDX , as considered the most sensitive to differences.

In most instances, as the reviewer foresaw, the $\%\Delta\text{HDX}$ significantly surpasses the $\%\Delta k_{\text{ch}}$, including for peptides spanning K417N (the table includes a 'note' column describing the outcome of the analysis). Therefore, while acknowledging the fact that the ΔHDX values have an offset at quantitative level, the observed ΔHDX can be considered qualitatively reliable. Only for one peptide (946-961 of delta spike), we cannot unambiguously demonstrate that the observed ΔHDX arises from a real difference in dynamics in respective to G614 spike. Therefore, we have not based any discussion on that peptide.

We now mentioned this in lines 184-187 of the results and included the description of this approach in the method session (lines 612-630), acknowledged the presence of this caveat. We also included new figures in the supplementary information (Fig. S4) where residue-level k_{ch} ratios are plotted.

“To compare peptides containing residue substitutions in spike variants (mutant peptides) with peptides of G614 spike, segments with identical N- and C-termini were selected. Their difference in deuterium incorporation (ΔHDX) was calculated according the equation 1 and plotted in Supplementary Figs. 9-12:

$$\Delta\text{HDX} = \left(\frac{\text{DU mutant peptide}}{\text{DU MaxD mutant peptide}} \times \text{DU MaxD G614 spike peptide} \right) - \text{DU G614 spike peptide} \quad (1)$$

To estimate the impact of the difference in chemical exchange rate constants (k_{ch}) on the observed ΔHDX between mutant peptides and peptides of G614 spike⁵⁰, firstly the k_{ch} of individual residues within the spike protein sequences were calculate⁴⁹. Successively, at peptide level, the percentage of difference in k_{ch} ($\%\Delta k_{\text{ch}}$) were compared to the percentage of ΔHDX normalized by the MaxD ($\%\Delta\text{HDX}$) in the time point showing maximal effect, as reported in Supplementary Data 1. The identified differences in HDX between spike variants and G614 spike in segments spanning amino acid changes resulted of high-confidence, with the impact of k_{ch} negligible, except for peptide 946-961 of delta spike.”

Supplementary Fig. 4. Influence of amino acid changes on the k_{ch} of residues of spike variants. The ratios between the k_{ch} of spike variants and Wuhan spike and the k_{ch} of G614 spike residues is plotted from residue 1 to 628 (a) and from residue 629 to 1256 (b). Values are extracted from Supplementary Data 1. Amino acid changes are illustrated on the left of the graphs.

Reviewer #3 (Remarks to the Author):

The authors have addressed many of the questions I had, and I commend them on including much of the data that is used to inform how they interpret their bimodal spectra. Based on this data and their response, there are a few last minor points that should be addressed prior to publication.

I agree with the authors that the isolated RBD experiment sheds a lot of light on the source of the bimodal spectra, so it likely does not stem from a lobe of the trimer simply being unbound. In line 322, the authors state “whilst the more exchanged (high-mass) population likely corresponding to a closed protomer transitioning to the open state and readying to engage another receptor molecule”. This statement makes it sound like the high-mass population should be able to bind Ace2, but then why is it unable to bind another molecule of Ace2? Maybe rephrasing this line to indicate that the high-mass population is somehow perturbed, but somehow still does not engage Ace2 like the other lobes of the trimer would help minimize confusion. If prior literature has made speculations about what this third lobe could be doing, then I recommend referencing those here.

- The attached additional spectra help confirm the reproducibility of the observed bimodals, but for several cases the two populations are so poorly resolved that deconvolution to extract data specific to each subpopulation can be ambiguous and potentially misleading. For example in figure S45 the spectra for G614 at 15 s and 1 min can likely be fit just as well with many other combinations of deuterium levels and intensities. The specific phrase: “The HDX of the high-mass population in the bound states appeared inconsistent with the HDX of the respective apo states.” should be clarified so that it specifically refers to the earliest time point where no evidence was seen for a population consistent with unbound RBD.

- In light of the attached spectra data there is one other possibility the authors should consider as a source of the observed bimodal spectra. Dissociation of ACE2 during deuterium exchange might also explain observed bimodals presented in Fig S42, S44, as unbinding of ACE2 during D2O incubation will likely start to occur in a matter of minutes. The general trend from Wuhan/G14 to Beta to Alpha showing later transitions in the EX1, appear to match the same trend in koff kinetics reported by Wrobel et al: <https://www.nature.com/articles/s41467-022-28768-w#MOESM1>. This source of bimodal would also be consistent with the earliest time point in the Ace2 bound form not yet showing a second (highly deuterated population) as the dissociation has yet to occur to any appreciable degree. I don't think the authors need to elaborate on this, but I think this is something the authors should at least mention as another possible confounding factor that is influencing the observed bimodal spectra in either the results or discussion.

- The authors should also include what temperature the pre-binding with Ace2 for 1 hour was conducted at. I know this seems nitpicky but there several labs working on similar systems and temperature may drastically affect binding kinetics for anyone attempting to reproduce these studies.

Reviewer #3 (Remarks to the Author):

The authors have addressed many of the questions I had, and I commend them on including much of the data that is used to inform how they interpret their bimodal spectra. Based on this data and their response, there are a few last minor points that should be addressed prior to publication.

I agree with the authors that the isolated RBD experiment sheds a lot of light on the source of the bimodal spectra, so it likely does not stem from a lobe of the trimer simply being unbound. In line 322, the authors state “whilst the more exchanged (high-mass) population likely corresponding to a closed protomer transitioning to the open state and readying to engage another receptor molecule”. This statement makes it sound like the high-mass population should be able to bind Ace2, but then why is it unable to bind another molecule of Ace2? Maybe rephrasing this line to indicate that the high-mass population is somehow perturbed, but somehow still does not engage Ace2 like the other lobes of the trimer would help minimize confusion. If prior literature has made speculations about what this third lobe could be doing, then I recommend referencing those here.

Thank you. We welcome the opportunity to better elaborate and agree with the reviewer: the high-mass population likely represents an unbound state, which is no longer closed but not properly open. Disordered RBD lobes characterizing such intermediate states have been described for cryo-EM datasets of SARS-CoV-2 spikes^{1,2,3}. In cryo-EM, these exposed RBD lobes generally constitute a minor population, as also seen by our HDX-MS analysis, and have been proposed to represent transient dynamic conformations leading to a fully open, ACE2-binding-competent form. This can explain why the HDX of high-mass population, which may be associated with these disordered RBDs, does not align to the low-mass population of the bound states or to the population of the unbound states. We have now rephrased the relevant passage referencing the cryo-EM studies^{1,2,3} (lines 313-314 and 323-326).

1. SARS-CoV-2 and bat RaTG13 spike glycoprotein structures inform on virus evolution and furin-cleavage effects. doi: 10.1038/s41594-020-0468-7
2. Distinct conformational states of SARS-CoV-2 spike protein. doi: 10.1126/science.abd4251
3. Structures and distributions of SARS-CoV-2 spike proteins on intact virions. doi: 10.1038/s41586-020-2665-2

- The attached additional spectra help confirm the reproducibility of the observed bimodals, but for several cases the two populations are so poorly resolved that deconvolution to extract data specific to each subpopulation can be ambiguous and potentially misleading. For example in figure S45 the spectra for G614 at 15 s and 1 min can likely be fit just as well with many other combinations of deuterium levels and intensities. The specific phrase: “The HDX of the high-mass population in the bound states appeared inconsistent with the HDX of the respective apo states.” should be clarified so that it specifically refers to the earliest time point where no evidence was seen for a population consistent with unbound RBD.

Following the reviewer’s advices, we have clarified that no evidence of a population consistent with unbound RBM is seen at the early HDX time points (lines 316-317). We have now clearly specified that the exact relative proportion of the low- and high-mass subpopulations cannot be derived from our data (lines 330-332) and modified the caption of figs. 5, S42 and S44 to clarify that our data need to be regarded as trends.

- In light of the attached spectra data there is one other possibility the authors should consider as a source of the observed bimodal spectra. Dissociation of ACE2 during deuterium exchange might also explain observed bimodals presented in Fig S42, S44, as unbinding of ACE2 during D2O incubation will likely start to occur in a matter of minutes. The general trend from Wuhan/G14 to Beta to Alpha showing later transitions in the EX1, appear to match the same trend in koff kinetics reported by Wrobel et al: <https://www.nature.com/articles/s41467-022-28768-w#MOESM1>. This source of bimodal would also be consistent with the earliest time point in the Ace2 bound form not yet showing a second (highly deuterated population) as the dissociation has yet to occur to any appreciable degree. I don't think the authors need to elaborate on this, but I think this is something the authors should at least mention as another possible confounding factor that is influencing the observed bimodal spectra in either the results or discussion.

We agree with the reviewer that the k_{off} of the complexes suggest that receptor dissociation starts occurring during the exchange reaction. The control experiment on the isolated RBD bound to ACE2, where a significant fraction of RBD remains unbound, indicates that mixture of bound and unbound RBDs does not manifest with a clear bimodality under the HDX conditions employed. In the same way, we believe it is unlikely that the spike RBDs remaining transiently unbound because of receptor dissociation could give rise to bimodality. However, following the reviewer's observation, we have adapted the text to include this alternative explanation and the evidence in support of this hypothesis (lines 313-318).

The whole paragraph now reads:

"...The more exchanged (high-mass) population represents RBDs perturbed by the presence of ACE2 but likely unbound: either because ACE2 transiently dissociates from them over the course of the exchange reaction or because they assume an intermediate, not fully erect state. The former hypothesis is supported by the observation that no evidence of a population consistent with the HDX of the unbound RBMs appears in the spike bound states at the early time points. However, the ACE2-bound state of the isolated ancestral RBD (3:2 RBD: ACE2), containing a significant fraction (33%) of unbound population, does not display bimodal isotopic distributions in the RBM under the conditions studied (Supplementary Fig. 41), suggesting that a simple mixture of bound and unbound populations of RBDs, even in the context of a spike trimer, would not manifest with an HDX bimodality either. Hence, we associated the high-mass populations of the bimodal HDX profiles with RBDs in an intermediate state, between closed and fully erect, receptor-binding-competent conformations. Such populations, characterized by disordered RBDs, have been observed and described before in cryo-EM studies^{11,15,55}. We thus rationalize, also based on the spike receptor binding mode reported in previous studies^{14,56}, that this population reports on the trimer capability to erect additional RBDs upon ACE2 binding to the neighbouring one/-s: a sign of cooperative opening. The exact relative proportion of the low- and high-mass subpopulations cannot be derived from our data as we cannot accurately deconvolve the two isotopic distributions. Nevertheless, the apparent abundance of the high-mass population seems to correlate with the overall preference of a given spike to adopt the open conformation as described above."

- The authors should also include what temperature the pre-binding with Ace2 for 1 hour was conducted at. I know this seems nitpicky but there several labs working on similar systems and temperature may drastically affect binding kinetics for anyone attempting to reproduce these studies.

We have now added this information (lines 577-579); the method session now reads:

“Before initiating the exchange reactions, spikes and ACE2 were incubated alone or in complex at ratio 1:2 spike trimer: ACE2 for one hour at the selected labelling temperatures (i.e. on ice or in the thermomixer at 23 °C or 28 °C) and the labelling buffer was as well temperature equilibrated.”